# An Overview of the Historical Retrofitting Interventions on Churches in Central Italy

Giorgia Cianchino [1], Maria Giovanna Masciotta [1], Clara Verazzo [2] and Giuseppe Brando [1,*]

1   Department of Engineering and Geology (INGEO), University "G. D'Annunzio" of Chieti-Pescara, Viale Pindaro 42, 65127 Pescara, Italy
2   Department of Architecture (DdA), University "G. D'Annunzio" of Chieti-Pescara, Viale Pindaro 42, 65127 Pescara, Italy
*   Correspondence: giuseppe.brando@unich.it; Tel.: +39 085-4537071

**Abstract:** The seismic sequence occurring in Central Italy in 2016 represents a new test benchmark for historical masonry churches and a chance for a better comprehension of their structural behavior under earthquake actions. The many earthquakes that took place in the past have led to stratifications of repair and retrofitting interventions that sometimes worsened the structural behavior, especially when resulting in the introduction of elements not compatible with the churches' original layout. Within this framework, the present paper intends to provide a critical review of the main interventions carried out in the churches of Central Italy for mitigating their seismic vulnerability and to evaluate their effectiveness in light of the damage surveyed on a representative sample of masonry churches after the 2016–2017 seismic sequence. The work is organized into three parts: (1) historical analysis of the territory; (2) review of the featured interventions; (3) critical appraisal of the interventions in relation to the surveyed data and assessment of their effectiveness. The goal of the work is to shed light on the correct design of retrofitting interventions in ancient masonry structures in order to enhance the structural safety of such artefacts without compromising their historical and cultural value.

**Keywords:** cultural heritage; masonry churches; historical interventions; seismic retrofitting; Central Italy; seismic damage





## 1. Introduction

Disaster recovery planning after extreme seismic events represents a crucial phase and a moment of reflection. Decisions made during the management of major emergencies must be timely and effective in both process and outcome [1]. In this context, particular attention must be provided when dealing with the seismic protection of cultural heritage (CH) [2,3]. The peculiar character and intangible value of CH structures make them unique objects, whose correct assessment encounters several challenges due to their geometrical complexity, the heterogeneity of constituent materials, the construction process, often stretched for many centuries, the adopted building techniques, as well as the presence of non-coeval interventions aimed at enhancing the structural behavior against non-ordinary actions [4,5]. A limited knowledge of the construction history and past events/alterations that ancient buildings have overgone over time can adversely affect a full understanding of their present condition [6]. This stresses the importance of a thorough knowledge-based approach to the preservation and seismic risk reduction of existing artefacts [7]. In this process, focus must be given to the historical evolution and stratifications featured in each structure, to the seismic hazard of the regions in which the heritage structure exists, and to the territorial and cultural identity of such areas.

It is worth stressing that the design processes of historical constructions did not follow codified technical standards; they are rather the result of "rudimentary scientific

approaches" [8], essentially empirical and rooted in lessons learned from the past, which allowed ancient builders to intuitively understand the static behavior of the structures and to adopt the most suitable construction solutions based on simple geometrical considerations (rules of thumb). Such an approach gradually brought the development of local construction techniques connected to the structural and architectural features of the buildings themselves [9–11]. Only during the 17th century this so-called "*Scientia Abscondita*" [12], handed in secrecy and passed down from generation to generation, was replaced by new procedures, also based on experiments, such as the analysis of the material strength of entire structural elements. In the light of the above considerations, it is clear that a critical historical analysis supported by a deep knowledge of both the construction methods and the intrinsic features of ancient structures is essential to obtain a reliable assessment of their seismic vulnerability and to design effective retrofitting solutions, thus avoiding inappropriate interventions driven by incompatible techniques.

Indeed, as reported in the literature, many retrofitting interventions carried out in the past were not beneficial in protecting against earthquake actions; on the contrary, they often led to significant faults and irreversible damage [13,14]. In Italy, the occurrence of various earthquakes within a relatively short period of time, e.g., Friuli 1980 [15], Umbria-Marche 1997 [16,17], Molise 2002 [18], Emilia 2012 [19], and Abruzzo 2009 [20–26], allowed observation of the seismic behavior of historical masonry churches and a better understanding of their fragility in order to assess cumulative damage. The last seismic event of 2016 further spotlighted the high vulnerability of these artefacts in Central Italy [27–29], enabling a deepening of the knowledge about the seismic response of masonry churches through the direct analysis of the surveyed damage, crack patterns and collapse mechanisms [30–33]. Rooted in the experience acquired during these events, new intervention techniques are being studied to find seismic upgrade solutions compatible with ancient masonry [34–37] and to understand whether or not these techniques will effectively make buildings safer and less prone to damage during future earthquakes. Recent experimental results from the scientific literature reveal, for instance, that GFRP or CFRP reinforcements provide cost-efficient solutions to improve the response of masonry structures under seismic actions [38,39], as these systems can inhibit both wall overturning and horizontal bending in nave walls, favoring a box behavior. Solutions based on joint reinforcement with steel bars, particularly applicable to masonry with regular courses, have been proposed as well [40]. Regarding roofing interventions, different diaphragm solutions, either rigid [41,42] or deformable [43], have been investigated in order to mitigate wall overloading. In this case, particular attention has been paid to evaluating the beam–masonry connection to prevent out-of-plane failure modes [44]. Common to all these interventions and improvement strategies is the cost optimization, aimed at justifying the economic investment driving the addressed solutions [45,46].

As for churches, retrofitting interventions are still a hot topic, as these buildings represent the layered historical identity of the Italian cultural heritage and are subject to many restrictions. As such, it is important to analyze thoroughly their seismic performance in order to carry out targeted interventions, capable of reducing their vulnerability [47] while preserving their cultural, artistic and historical value. To this end, this paper presents an overview of the most widespread anti-seismic interventions conducted in Italian churches and provides a critical analysis of their effectiveness in light of the 2016 Central Italy earthquake.

The study stems from an in-depth reconnaissance activity carried out on behalf of the Department of Civil Protection in support of the Italian Ministry of Cultural Heritage, which involved various researchers from several Italian Universities, who shared photos, evaluation reports and other documents, such as the compiled A-DC forms for churches [48]. The data collected were made available to the scientific community through the development of the Da.D.O. (Observed Damage Database) web platform, developed by the Eucentre Foundation for the Italian Civil Protection [49]. In this platform, 4016 churches—surveyed for emergency management purposes after the 2016–2017 seismic sequence—were cata-

logued in terms of structural/stylistic features, damage and photos. Almost 600 of these churches were subjected to retrofitting interventions in the past, 145 of which were directly surveyed by the authors (Figure 1).

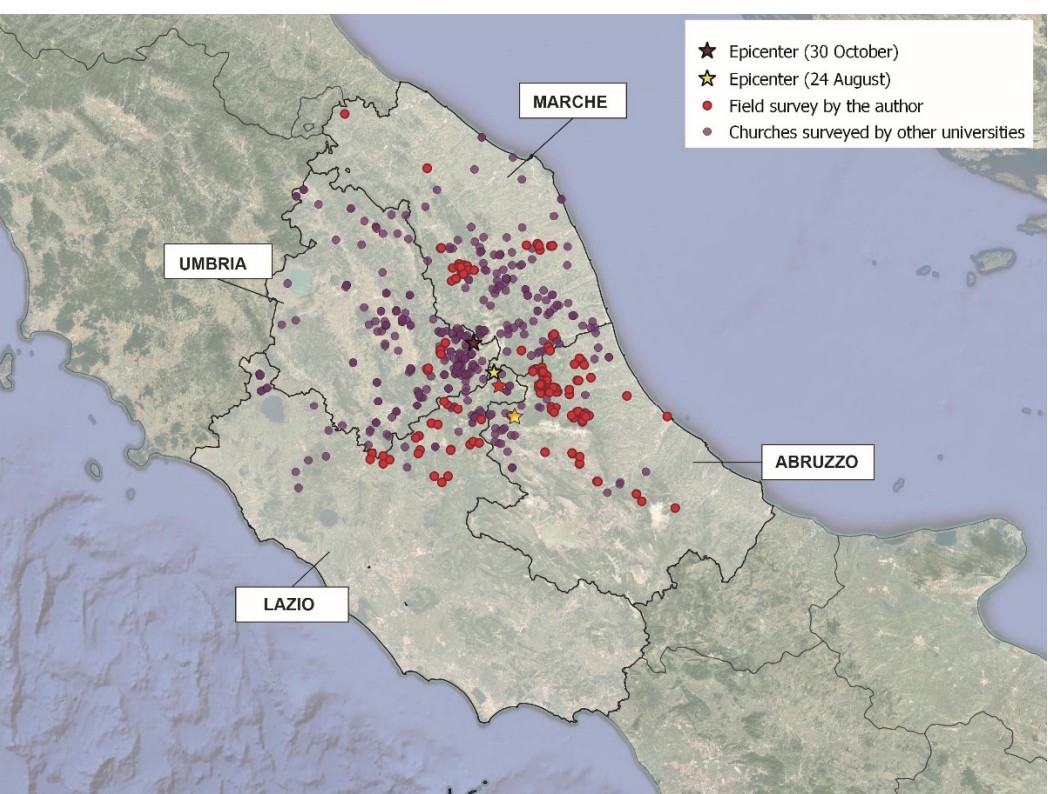

**Figure 1.** Location of analyzed churches in Central Italy: distribution in the regions of Umbria, Marche, Eastern/Northern Lazio and Northern Abruzzo.

The considered sample of 600 churches is spread throughout a large territorial area that has a fair seismic history (the most recent event before 2016 was the Umbria-Marche 1997 earthquake), characterized by earthquakes with different magnitudes and zones that were differently affected. This allowed us to observe several levels of intervention on the original structures and to identify the post-earthquake reconstruction during which they were applied.

Several studies have been published in recent times about the effectiveness of antiseismic devices for ordinary masonry buildings during the 2016–2017 seismic sequence. In most cases, the analyses were limited to the municipal level; relevant examples in this regard are the works by Sisti et al. [50] and Trifan et al. [51], to mention a few, with both having as the object of study the buildings located within the historical hearth of Norcia, the town closest to the epicenter of the October 2016 earthquake.

Regarding Central Italy masonry churches, many scientific works have addressed their structural performance under seismic actions, yet no systematic correlation between damage levels and past interventions has been ever drawn. Moreover, the focus has been on specific churches or to a circumscribed number of them [52–55], hindering statistically significant analyses at the regional and supraregional scales. Conversely, thanks to the extensive information collected through the aforementioned activities, this study provides a methodical review of the historical interventions carried out over time in the masonry churches of Central Italy along with a statistical analysis of the effects that these exerted on the seismic performance of a considerable number of churches struck by the earthquakes in 2016–2017.

Considering the historical and cultural importance of these structures, the knowledge acquired through the present work can be of great help in prioritizing cost-effective retrofitting measures and driving large-scale seismic risk mitigation strategies.

The paper is structured as follows. Section 2 describes the historical context characterizing the territory of Central Italy to allow an understanding of the political and architectural choices driving past interventions on CH assets. Section 3 provides a critical analysis of the main retrofitting measures carried out in the masonry churches located across the investigated area. Section 4 discusses the effectiveness of such interventions based on the damage surveyed on a selected sample of churches after the seismic sequence that struck Central Italy in 2016; to this purpose, the data directly managed by the authors were integrated with the data extracted from the Da.D.O. platform. Finally, Section 5 concludes the work.

## 2. Churches in Central Italy

### 2.1. Historical and Political Framework

The churches of Central Italy, as they appear today, are the direct outcome of the political context and developments in the country over the centuries. The Papal State, which ruled several territories in Central Italy for over a millennium until 1870, wanted to keep these areas divided into very small districts in order to exercise better control [56]. At the same time, the need to promote as much as possible the Catholic religion led to an immeasurable growth of churches. As such, the small municipalities that fall within the regions of Umbria, Marche, Eastern Lazio and Abruzzo nowadays present a large plethora of valuable religious buildings that need special attention and care, as they belong to a territory characterized by recurrent seismicity.

In particular, compared to more densely populated regions of Northern Italy (Figure 2), regions in Central Italy present a larger number of churches (>200), whose density is not directly correlated to the size of the municipalities, resulting in analogous distributions regardless of the population density (source: GeoNue by Nordai Srl). Many of these churches were built—or rebuilt after earthquakes—according to the political decisions of the papacies to control the territories in terms of population and social and economic activities. In many cases, the principal drive behind the papacy choices upon its long-standing territories was the economic profit [57]. This modus operandi inevitably impacted the coeval architecture of churches, which suffered both successes and failures of the Papal power over centuries.

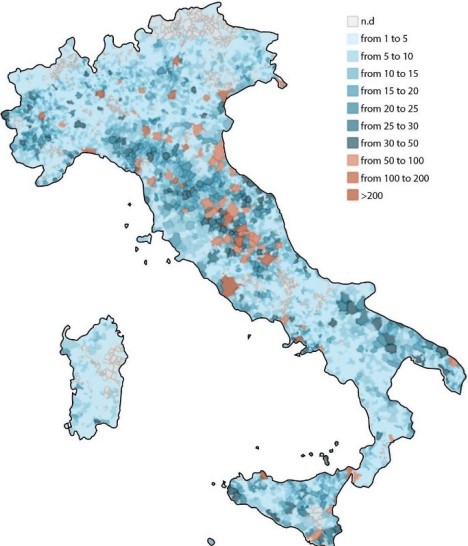

**Figure 2.** Italian churches: distribution of the number of churches by municipality (created with GeoNue by Nordai Srl: https://www.geonue.com/le-chiese-in-italia, accessed on 18 July 2022).

### 2.2. Geometric and Stylistic Characteristics

#### 2.2.1. Influence of the Mendicant Orders

Mendicant orders played a decisive role in the approach to sacred spaces, significantly influencing the architecture of the churches in Central Italy. At the beginning of the XIII century, two groups of Christian religious orders, the Franciscans and the Dominicans [58–60], committed to returning to the pure spirituality of the Christian Church, establishing and spreading throughout the Italian Peninsula. They arose with the aim of promoting an ascetic way of life dedicated to prayer and poverty, as opposed to the corruption of the Church. The concepts of poverty and simplicity, along with the refusal of the mendicants to own property, were unavoidably reflected on the architectural structures of the new orders.

Initially settled in small and abandoned churches, the mendicant friars rapidly spread in the poorer towns and cities of Italy, giving rise to the first convents. These buildings were usually located in areas outside the city but still at the intersections of the main streets. However, the need for a straightforward communication with the faithful entailed the creation of settlements in other cities and towns. Mendicant architecture was intimately tied to the principle of poverty; thus, the first churches were very simple, featuring a single bare nave covered with trussed wooden roofs and gabled facades.

#### 2.2.2. Stylistic and Spatial Changes

Afterwards, the "monumentalization" of the building took over, paving the way for the second phase of the mendicant architecture [56], during which church layouts became more complex. Meanwhile, the French influences reached Central Italy, specifically the Umbria region [61], bringing new construction styles. In the richer cities, mendicant church architecture started to be permeated by stylistic changes in the typology, leading progressively to the Italian Gothic style. Ribbed vaulted systems (first derived from the rounded arch form, then from the ogival arch form) replaced the typical timber roof trusses used to bridge the space above the naves, becoming one of the most distinctive features of the Gothic architecture, along with flying buttresses. As time passed, the mendicant orders lost their initial obsession with poverty, and thus more sophisticated stylistic models begun to be accepted. Figure 3 provides a schematic insight into the evolution of the main structural characteristics of mendicant churches over time.

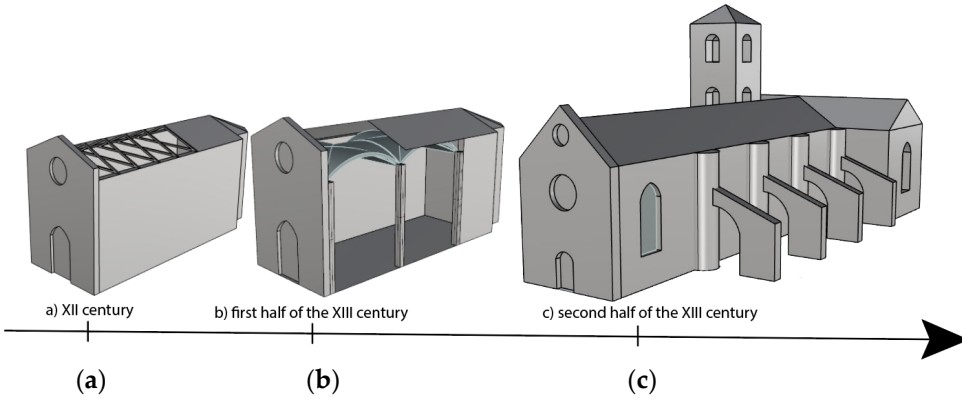

**Figure 3.** Stylistic evolution of mendicant churches over the medieval age: (**a**) Single nave church covered by wooden trusses; (**b**) Single nave church covered with ribbed vaults; (**c**) Monumentalization of the church in the most important cities (the French influence emerges from the use of buttresses and pointed arches).

The inclusion of churches within urban centers also influenced their planimetric development. In this regard, thanks to the Church Database developed during the field reconnaissance activities dealt with in this paper [62], it was possible to analyze the geometrical features of about 600 churches located in Central Italy—hereafter referred as the sample—and to compare their planimetric dimensions against their construction period. It was found that the churches dating back to the medieval age present a preponderant

longitudinal development, while churches built afterwards feature a more compact plan. This is likely due to the fact that the churches built from the Renaissance onwards were often located in the center of the towns; hence, their architectural design was directly conditioned by the urban layout, resulting in buildings with reduced proportions. The remarks reported above are synthesized in Figure 4a, where the highlighted percentages refer to the distribution of the considered sample within each construction period. It is noted that most of the churches (33%) were built between the 15th and the 17th centuries. Overall, the average aspect ratio (length to width, $L_1/W_1$) ranged from 2.1 in the 9th century to 1.7 in the 20th century. As an example, Figure 4b,c shows the planimetric scheme of two churches located in Central Italy (Umbria region) but belonging to different historical periods: San Francesco in San Gemini (13th century) and San Nicola in Monteleone di Spoleto (18th century).

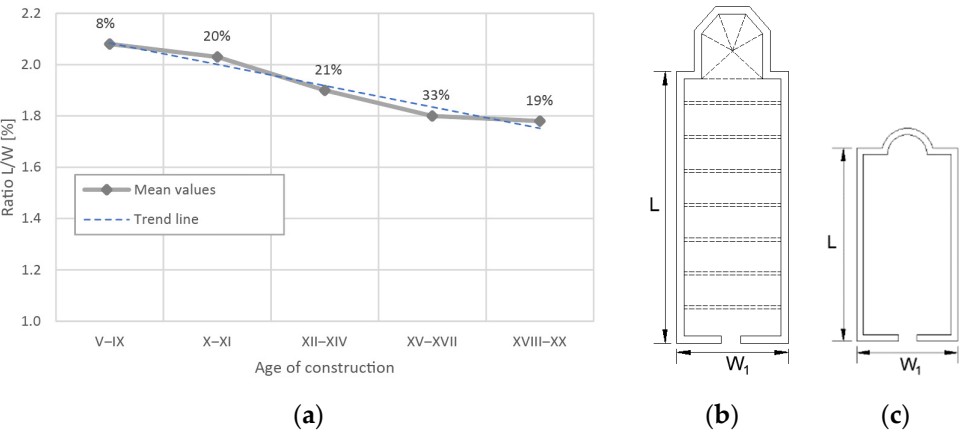

(a)            (b)       (c)

**Figure 4.** Planimetric development of Central Italian churches over centuries: (**a**) Length to width ratio versus age of construction; (**b**) Scheme of San Francesco floor plan (San Gemini-Terni); (**c**) Scheme of San Nicola floor plan (Monteleone di Spoleto). Percentages refer to the distribution of the entire sample (600 churches) within each construction period.

Analogous considerations emerged by analyzing the same sample of churches with respect to the proportions of their facade. In fact, their aspect ratio (height to width $H/W_2$) changed over time, passing from stocky configurations in the early centuries ($H/W_2 < 1$) to churches with slender facades between the 12th and the 17th centuries ($H/W_2 > 1$) (Figure 5).

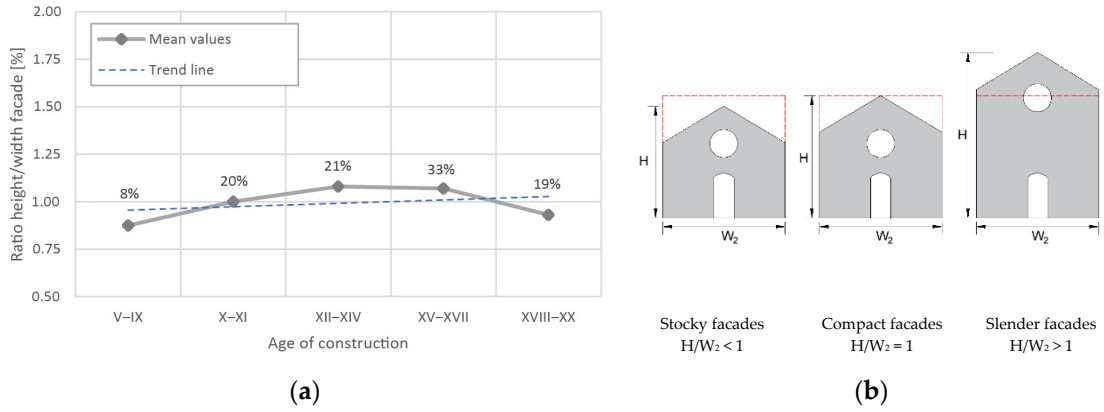

(a)            (b)

**Figure 5.** Aspect ratio evolution of the main facades of Central Italian churches: (**a**) Height to width ratio versus age of construction; (**b**) Facade proportion against a superimposed quadrilateral with aspect ratio equal to one. Percentages refer to the distribution of the entire sample (600 churches) within each construction period.

Furthermore, the adoption of new construction systems and shapes, such as pointed arches [63] and derived vaulted systems, progressively led to lighter structures, with perimetral walls becoming slimmer and slimmer.

Indeed, ogival arches made possible to reach greater heights, to decrease the arch lateral thrust and to direct the self-weight of the vault to the four corner pillars, thus avoiding the wide supports or massive buttressing systems, which were typical of the Romanesque period and allowed lateral walls to be reduced in thickness [64], leaving room for large stained glass windows (Figure 6). Variable geometric factors, such as thickness, span and radius, greatly influenced the behavior of vaulted systems and, consequently, the structural response [65].

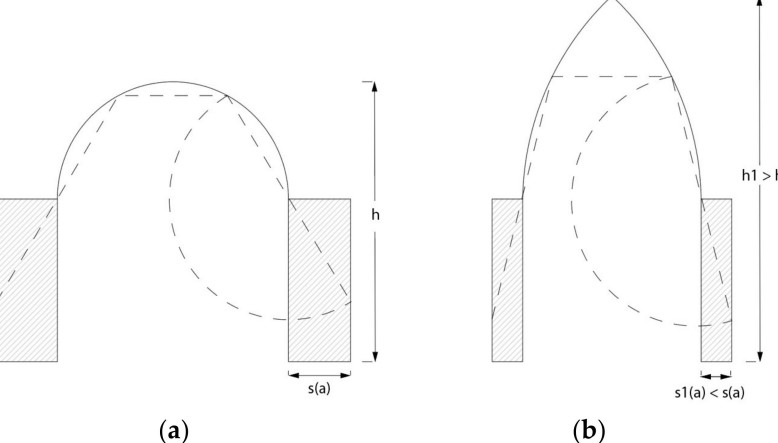

**Figure 6.** Evolution of the arch system and spatiality of churches: (**a**) Romanesque round arches and massive masonry; (**b**) Gothic pointed arches and slender walls.

Stylistic changes deriving from the progressive adaptation of the churches to new styles, and sometimes to new spatial needs, were also considered with the aim of analyzing the influence that these alterations had on the structural stability of the investigated sample. The highly decorative and theatrical dictates of the new design provided very rich painted stuccoes and sophisticated roofing, making the churches more contemporary. The church of San Domenico in Spoleto, in the Umbria region, is a representative example of this stylistic transformation process, which first saw the incorporation of Baroque decorations in the interior spaces (Figure 7a) and then their removal in order to return the inner sacred space to its original configuration (Figure 7b,c).

Antithetical to these spatial changes are indeed the changes that occurred in other churches of nearby areas, such as the church of Madonna della Pietà in the Province of Teramo (Abruzzi region), more than 100 km away from the epicenters of the last earthquake of 2016. Here, past spatial alterations were mainly driven by functional reasons and often resulted in the closure of existing openings (Figure 8a). The visual inspection of these posterior additions revealed critical aspects either associated with the poor quality of the masonry material used for the purpose or related to the different shape, size and arrangement of the employed masonry units. For instance, openings within regular stonemasonry walls with brick courses were often closed off by filling them with irregular stones and rubble material, thus producing local discontinuities and separation cracks, as the one that appeared at the arch intrados of the walled-up doorway in San Francesco Church in Cascia (Figure 8b).

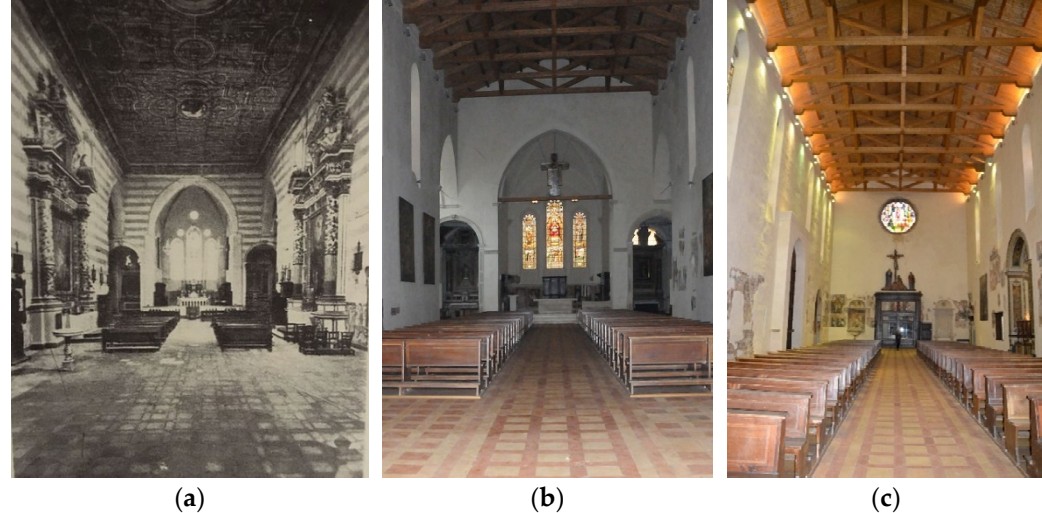

**Figure 7.** Interior of San Domenico Church in Spoleto: (**a**) Baroque style of 1600 apse (source: Wikipedia); Restoration of the original style since 1934: (**b**) apse view (source: Wikipedia) and (**c**) facade view.

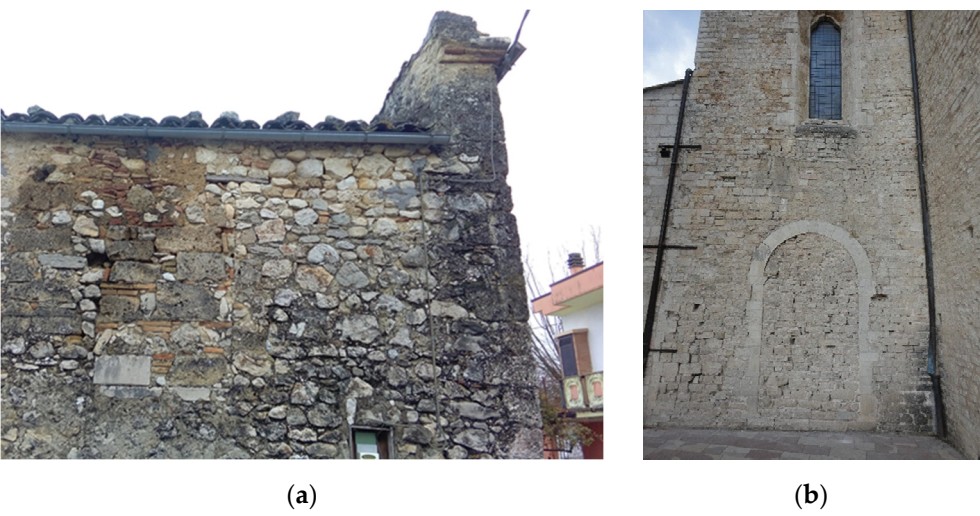

**Figure 8.** (**a**) Madonna della Pietà Church—Isola del Gran Sasso (TE); (**b**) S. Francesco Church—Cascia (PG).

### 2.3. Seismic History and Churches Post-Earthquake Reconstructions

A visual insight into the strongest earthquakes occurring in the territories of Central Italy from the 17th century onwards is provided in Figure 9. According to their magnitude, three types of earthquakes can be distinguished: "small" ($4.0 \leq Mw < 5.0$), "medium" ($5.0 \leq Mw < 6.0$) and "large" ($Mw \geq 6.0$). The strong seismicity of the area [66], characterized by repeated "medium" and "large" earthquakes within a relatively short time span, led to several reconstructions closely spaced in time [67].

The 17th century was characterized by many "medium" earthquakes (Figure 9a); in addition to these seismic disasters, the period was marked by the outbreak of many epidemics. Thus, the Popes' efforts were mainly focused on healing the economic crisis and stopping the depopulation of the cities rather than planning proper reconstruction measures for damaged artefacts.

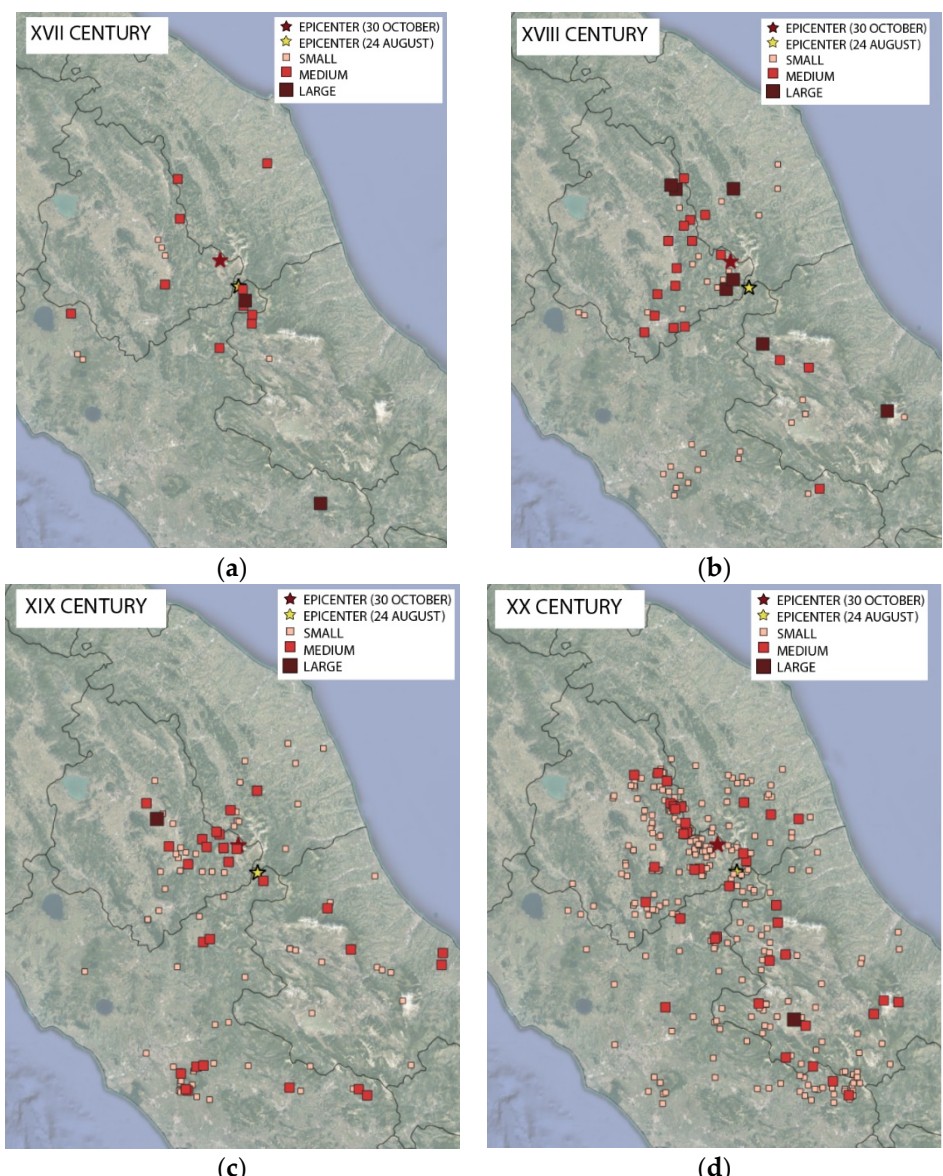

**Figure 9.** Earthquakes in Central Italy during the centuries: (**a**) 17th century; (**b**) 18th century; (**c**) 19th century; (**d**) 20th century. "Small" earthquakes ($4.0 \leq Mw < 5.0$) are highlighted in orange, while "medium" ($5.0 \leq Mw < 6.0$) and "large" ($Mw \geq 6.0$) earthquakes are marked by light and dark red squares, respectively (data retrieved by ASMI—Italian Archive of Historical Earthquake Date: https://emidius.mi.ingv.it/ASMI/, accessed on 28 January 2021).

As a result, the post-earthquake restoration was highly extemporaneous. Some religious buildings were even rebuilt on the ruins of destroyed churches; however, a growing awareness of the improper use of repair materials and techniques already existed at that time, as documented in some of the technical reports written by the Pope's engineers and experts [57].

Numerous "large" earthquakes occurred instead during the 18th century (Figure 9b). Among them, the 1703 Apennine earthquakes (6.7 Mw) that struck L'Aquila and the area of Valnerina are worth being mentioned because of the significant number of casualties recorded as well as the extensive damage caused to cultural heritage buildings. As a result of these catastrophic telluric events, anti-seismic devices such as ties or buttresses gradually became part of the ordinary building practice, leading to the development of a local seismic culture aimed at increasing the resistance of masonry structures to earthquakes, in particular with respect to the out-of-plane mechanisms. Indeed, there has

been always a close correlation between the development of seismic-resistant building practices and earthquake frequency. If long periods of time pass between seismic events, the collective memory is lost, and the "culture of prevention" is gradually replaced by a "culture of repairs" [67]. The widespread employment of anti-seismic solutions after the 1703 earthquakes emerged during the surveys carried out after L'Aquila earthquake of 2009. Particularly, as far as the churches are concerned, wooden elements embedded in masonry walls were found as typical anti-seismic presidia with the function of ties. Analogous measures, probably already known in the Middle Ages as seen in the defensive architecture of the Abruzzo area [68], were also adopted for residential buildings, as further discussed in Section 3.1.

During the 19th century, no high-intensity earthquakes hit Central Italy (Figure 9c); hence, the empirical knowledge about seismic prevention that had been passed down orally over the centuries began to be forgotten and started to be replaced by "official" codified practices and common building regulations. A reduced use of anti-seismic devices for masonry churches is documented in this period [69].

Only in the 20th century did the Italian State manage to define post-earthquake reconstruction models. Although not very strong, the numerous seismic events that marked this period (Figure 9.d) became the pretext for experimenting with new building materials and construction techniques [70]. For the sake of clarity, Table 1 shows the location and magnitude of the most influential earthquakes, with Mw ≥ 5.0, that hit Central Italy over the centuries, highlighting those with Mw ≥ 6.

**Table 1.** Location and magnitude of the major earthquakes (Mw ≥ 5.0) that occurred in Central Italy over the last four centuries. Large earthquakes with Mw ≥ 6.0 are highlighted in dark red.

| Year | Epicentral Area | Mw | Year | Epicentral Area | Mw | Year | Epicentral Area | Mw |
|---|---|---|---|---|---|---|---|---|
| 1612 | Appennino umbro-marchigiano | 5.1 | 1804 | Gran Sasso | 5.4 | 1916 | Aquilano | 5.1 |
| 1619 | Aquilano | 5.3 | 1806 | Colli Albani | 5.6 | 1916 | Alto Reatino | 5.5 |
| 1626 | Macerata | 5.1 | 1815 | Valnerina | 5.6 | 1917 | Ternano | 5.0 |
| 1627 | Monti della Laga | 5.3 | 1821 | Rieti | 5.1 | 1922 | Val Roveto | 5.2 |
| 1631 | Appennino umbro-marchigiano | 5.1 | 1832 | Valle Umbra | 6.4 | 1927 | Marsica | 5.2 |
| 1639 | Monti della Laga | 6.2 | 1832 | Appennino umbro-marchigiano | 5.4 | 1933 | Maiella | 5.9 |
| 1646 | Monti della Laga | 5.9 | 1838 | Valnerina | 5.1 | 1933 | Maiella | 5.1 |
| 1654 | Sorano | 6.3 | 1838 | Valnerina | 5.5 | 1941 | Monti Sibillini | 5.0 |
| 1667 | Spoleto | 5.1 | 1838 | Valnerina | 5.2 | 1943 | Monti Sibillini | 5.0 |
| 1672 | Monti della Laga | 5.3 | 1854 | Valle Umbra | 5.6 | 1943 | Ascolano | 5.7 |
| 1689 | Reatino | 5.1 | 1859 | Valnerina | 5.7 | 1948 | Monti Reatini | 5.4 |
| 1695 | Lazio settentrionale | 5.8 | 1873 | Appennino marchigiano | 5.9 | 1950 | Gran Sasso | 5.7 |
| 1703 | Valnerina | 6.9 | 1873 | Val Comino | 5.4 | 1951 | Gran Sasso | 5.3 |
| 1703 | Aquilano | 6.7 | 1874 | Aquilano | 5.1 | 1951 | Monti Sibillini | 5.3 |
| 1706 | Maiella | 6.8 | 1874 | Val Comino | 5.5 | 1958 | Aquilano | 5.0 |
| 1707 | Monti Martani | 5.2 | 1876 | Monti Prenestini | 5.1 | 1961 | Reatino | 5.1 |
| 1714 | Narni | 5.3 | 1877 | Lazio meridionale | 5.2 | 1962 | Valnerina | 5.0 |
| 1719 | Valnerina | 5.6 | 1878 | Valle Umbra | 5.5 | 1962 | Valle Umbra | 5.3 |
| 1721 | Appennino umbro-marchigiano | 5.1 | 1879 | Valnerina | 5.6 | 1972 | Marche meridionali | 5.5 |
| 1730 | Valnerina | 6.0 | 1881 | Chietino | 5.4 | 1973 | Valle del Chiascio | 5.1 |
| 1745 | Valle Umbra | 5.1 | 1882 | Chietino | 5.3 | 1979 | Valnerina | 5.8 |
| 1747 | Appennino umbro-marchigiano | 6.1 | 1882 | Costa ascolana | 5.2 | 1984 | Monti della Meta | 5.5 |
| 1747 | Appennino umbro-marchigiano | 5.4 | 1883 | Monti Prenestini | 5.1 | 1987 | Costa Marchigiana | 5.1 |
| 1751 | Ternano | 5.1 | 1883 | Monti della Laga | 5.1 | 1997 | Appennino umbro-marchigiano | 5.7 |
| 1751 | Appennino umbro-marchigiano | 6.4 | 1892 | Colli Albani | 5.1 | 1997 | Appennino umbro-marchigiano | 6.0 |
| 1762 | Aquilano | 5.5 | 1898 | Reatino | 5.5 | 1997 | Appennino umbro-marchigiano | 5.2 |

**Table 1.** *Cont.*

| Year | Epicentral Area | Mw | Year | Epicentral Area | Mw | Year | Epicentral Area | Mw |
|---|---|---|---|---|---|---|---|---|
| 1767 | Valle Umbra | 5.5 | 1898 | Valnerina | 5.0 | 1997 | Appennino umbro-marchigiano | 5.5 |
| 1771 | Sorano | 5.1 | 1898 | Valnerina | 5.5 | 1997 | Valnerina | 5.2 |
| 1785 | Appennino umbro-marchigiano | 5.1 | 1899 | Colli Albani | 5.1 | 1997 | Valnerina | 5.6 |
| 1785 | Monti Reatini | 5.8 | 1901 | Sabina | 5.3 | 1998 | Appennino umbro-marchigiano | 5.0 |
| 1791 | L'Aquila | 5.3 | 1901 | Sorano | 5.2 | 1998 | Appennino umbro-marchigiano | 5.3 |
| 1791 | Appennino umbro-marchigiano | 5.6 | 1904 | Marsica | 5.7 | 1998 | Appennino umbro-marchigiano | 5.1 |
| 1792 | Ternano | 5.1 | 1905 | Valle Peligna | 5.2 | | | |
| 1793 | Appennino umbro-marchigiano | 5.3 | 1915 | Marsica | 7.1 | | | |
| 1799 | Appennino marchigiano | 6.2 | 1915 | Marsica | 5.0 | | | |
| 1799 | Foligno | 5.1 | 1915 | Marsica | 5.1 | | | |

## 3. The Main Interventions Observed in Central Italy

As previously mentioned, the numerous earthquakes that repeatedly hit Central Italy over the centuries led to the development of a local building culture, resulting from the direct observation of the seismic damage, the consequent interpretation of the structural response, the identification of the most influencing fragility sources and the field experimentation of possible countermeasures. Anti-seismic devices have likely been used ever since, but their systematized and standardized employment as earthquake-resistant solutions was documented only starting from the 17th century, when a more scientific approach began to consolidate around the adopted solutions.

Aware that the lack of effective connection between structural elements and the poor quality of the material are the main causes of damage in masonry churches [71], traditional earthquake construction techniques employed by ancient masons were mostly aimed at improving these aspects [67,72] as well as reducing the thrusts of arches and vaults [73]. A detailed overview of the main anti-seismic solutions recurring in the churches in Central Italy is provided in the next sections.

### 3.1. Wooden Elements

One of the most consolidated "rules of thumb" to improve the seismic response of masonry churches under earthquakes was the insertion of wooden elements into walls and vaulted systems.

Different technical procedures were used to embed these strengthening devices within the masonry. The aim was to exploit the tensile strength of the timber so as to compensate for the poor tensile resistance of the masonry and to reduce sliding between rubble stones. Also known as "ties" or "roots", wooden elements were usually placed within the entire thickness of the wall running from one face to another, in order to bind together multiple leaves and improve the wall stability by preventing crack propagation (Figure 10a). Another way to apply these roots was to arrange them continuously within the perimeter of the building, at the floor level, holding the walls together and fostering a box-like behavior (Figure 10b). This latter timber-laced solution was a very common practice [69] in the seismic areas of Central Italy. An example found in the church of S. Biagio D'Amiterno, damaged by the 2009 Aquila earthquake, can be observed in Figure 11. It is worth noting that, besides exerting a reinforcement function, embedded wooden elements also helped prevent structural failures during the construction process of masonry churches, especially in the presence of vaults [74], domes and bell towers.

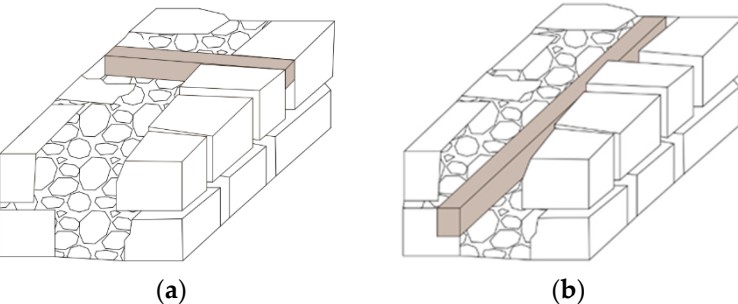

(**a**)     (**b**)

**Figure 10.** Insertion of wooden elements into walls: (**a**) transversal wall reinforcement; (**b**) longitudinal wall reinforcement (along the perimeter).

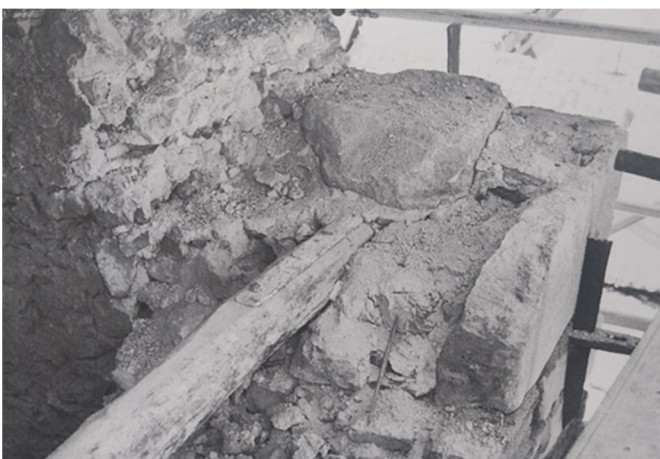

**Figure 11.** Wooden element inside the masonry of S. Biagio D'Amiterno in L'Aquila (source: [69]).

As far as vaulted systems are concerned, the insertion of wooden beams had the specific purpose of absorbing the horizontal thrust. Occasionally, not to be visible from the inside, these elements were positioned at the vault extrados, but their effectiveness in restraining thrust-induced outward movements at the springing level was limited (Figure 12a). Indeed, extrados tie-rods provide an efficient restraining action only in the presence of high compression loads, as vertical actions counteract the pier rotation (Figure 12b). The placing of ties in line with possible hinge locations was used in the past: in fact, specific tying devices called "tie slings" (*tiranti a braga*) were often used to transfer the effect of the tie-rods to the arch impost and to prevent the rotation of the piers. To make these anti-seismic devices further effective, they were flanked by two additional wooden beams counteracting the deformations of the vault (Figure 12c). This solution was adopted for instance in the vaults of the central nave of the church of San Domenico in L'Aquila (Figure 13a). It is noted that, to prevent the horizontal tie rod from deforming owing to the presence of oblique elements, the device must be suitably rigid. Still, this was not always the case in historic structures.

A different solution, which became the most popular, was the insertion of wooden beams at the level of the arch impost. Although visible from the interior of the church, this solution proved to be more efficient in counteracting the lateral thrust of the vaults.

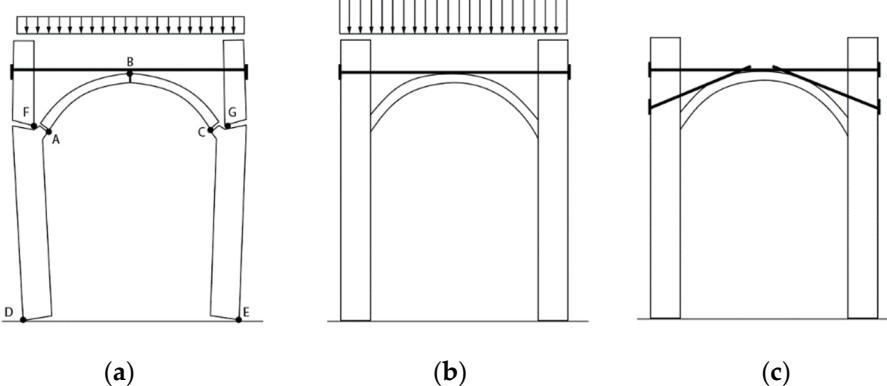

**Figure 12.** (**a**) Possible collapse mechanism in an arch-piers system with extrados ties; (**b**) containment of pier rotation due to high vertical actions; (**c**) insertion of oblique beams to contain the thrust of the vault and the rotation of the piers.

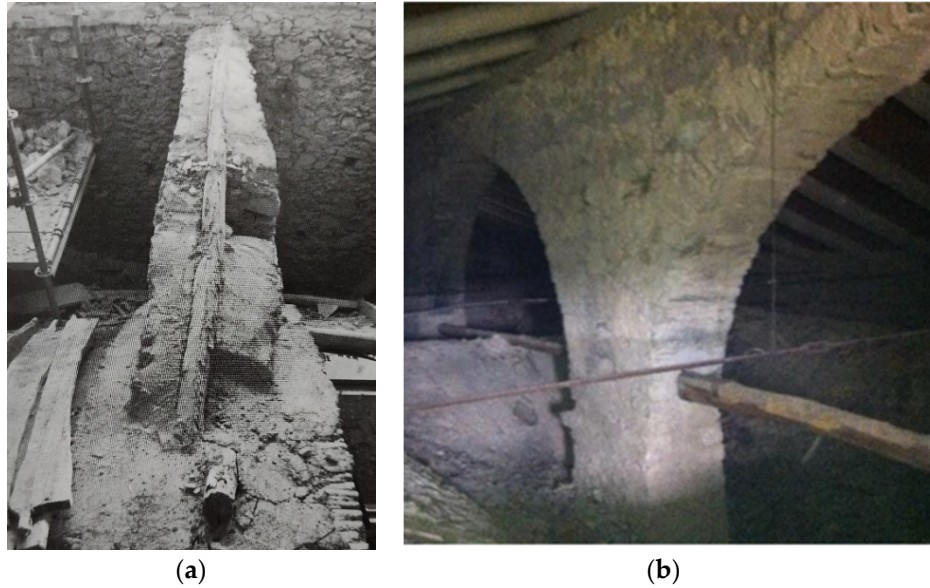

**Figure 13.** (**a**) Example of "tie slings" in the vaults of the church of San Domenico in L'Aquila; (**b**) Wooden tie in the attic of S. Andrea church in Campi (source: [75]).

Both the seismic sequence of 2016 in Central Italy and the earthquake of 2009 in L'Aquila brought to light the widespread use of wooden elements inside many churches, e.g., the church of Sant'Andrea in Campi (Figure 13b) [75]. Independently from the location, the major issue with this type of device was its slippage within the masonry. To work properly, ties have to be well fastened at the ends; the employment of anti-pull metal bars (lately replaced by anchor plates) represented a practical solution to solve this problem, preventing slipping and improving the wall connection. The anti-pull systems were nailed to the wooden tie rod and kept into place though a stake inserted into the end slot of the metal bar, whose shape also evolved with time from pointed to flat (Figure 14a,b).

*3.2. Metal Ties*

Due to the intrinsic risk of deterioration of organic materials, wooden elements were slowly replaced by metal ties [76,77]. Generally, the latter were not hidden inside masonry elements but visibly positioned both in the longitudinal and transversal directions of the structure, thus allowing them to be added a posteriori (namely after the construction phase), with a precise anti-seismic consciousness to inhibit collapse mechanisms.

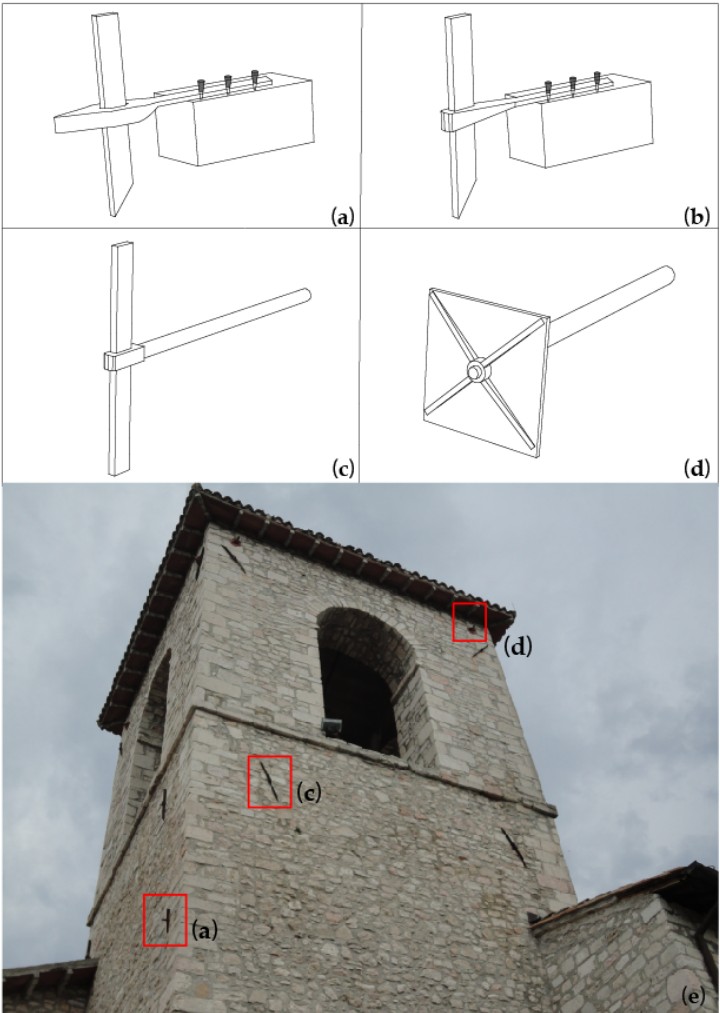

**Figure 14.** Evolution of anchor bars and plates in Central Italy. In the bell tower of Sant'Agostino church in Cascia, it is possible to notice various types of anchors inserted in different periods with the function of hoops: anti-pull systems with (a) pointed bar, (c) flat bar and (d) metal plates.

### 3.2.1. Evolution of Materials and Geometry

Initially made of forged iron and later replaced by steel or other alloys [78], metal ties have been used in Italy to strengthen masonry buildings since the 15th century. As for the territorial area of Central Italy, differences can be found in the geometry and shape of these devices; yet, the observed typical square or circular shapes of the cross section do not allow inference about the historical period to which they date back. On the contrary, the study of anchor bars and plates allows interesting cues. Older tie rods usually feature pointed ends, and the length of the anchor bar does not exceed 35 cm (Figure 14a). Representative examples can be found in the area of L'Aquila, where anchors with "lily-shaped" washers result as the most common tying solution. The increasing awareness about the advantages and limitations of these devices against earthquakes led to the preference of flat-end tie rods with anchor bars longer than 35 cm over the former alternative, aiming at avoiding punching failures because of the insufficient interlocking length between elements (Figure 14c).

The traditional practice of embedding tie rods inside the masonry thickness with the purpose of hiding them was strongly discouraged by important architecture masters of the time, such as *Valadier*, because it would have weakened the part of the masonry where the tie rod was acting, inducing significant stress concentrations. In the case of historical masonry composed of small-sized blocks, the contrasting effort exerted on the masonry

would not be entrusted to the small size of the anchorage. Therefore, in order to increase the contact surface between the anchor bar and the masonry, thus avoiding the punching mechanism, during the 20th century, the anchor bars frequently used in churches were replaced by metal plates (Figure 14d). An in-depth history of this evolution is provided in the book *"Ita Terraemotous Damna Impedire"* [48]. Figure 14e shows the application of some of the aforementioned types of anchors to the bell tower of the church of Sant'Agostino in Cascia, Perugia.

### 3.2.2. Placement

As for the position of tie rods with specific reference to the investigated sample of churches in Central Italy, the adopted layouts were always driven by the necessity to avoid the most likely potential collapse mechanisms. As an example, Figure 15a,b shows the typical location of longitudinal and transversal ties aimed at restraining perpendicular load-bearing walls, with the purpose of preventing their out-of-plane tilting under horizontal seismic forces.

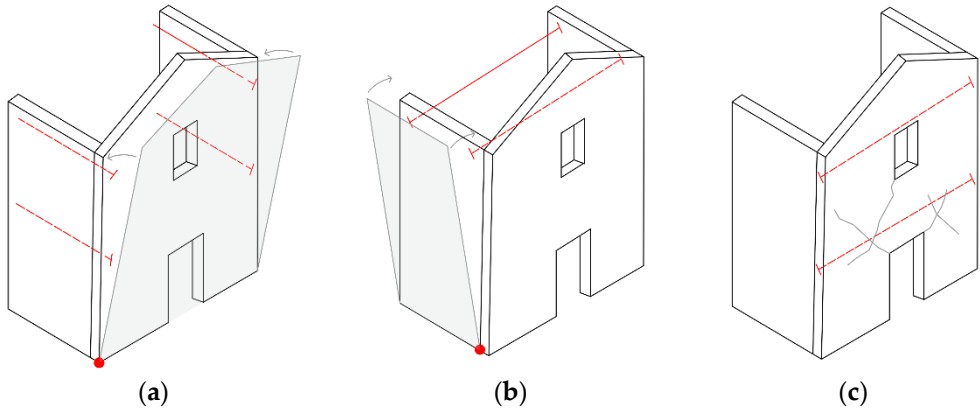

|     |     |     |
| :-: | :-: | :-: |
| (**a**) | (**b**) | (**c**) |

**Figure 15.** Positioning of the tie-rods: (**a**) longitudinal (parallel to the naves); (**b**) transversal (perpendicular to the naves); (**c**) in the counter-facade (in adherence to the wall thickness).

Although not visible from the internal room of the church, being often placed in adherence with the lateral walls and hidden by decorative frames—as in the case of Figure 15a—the presence of tie rods is always evident in the facade because of the external plates or bars generally positioned midway between half of the wall height and the top. Tie rods were also applied as a reinforcement measure to counteract in-plane mechanisms; in this case they are found in adherence to masonry, typically over the portal or rose window if the wall is a facade (Figure 15c). Metallic tie-rods stand as the most used anti-seismic devices for unreinforced masonry (URM) structures in Central Italy [79]. Some interesting examples of their use are given in Figure 16a,b: the first reports a lateral view of the church of S. Antonio in Cascia, where the presence of transversal and counter-facade ties is clearly visible; the latter is a perspective view of the facade of church of S. Antonio in Norcia, which allows a view of both the longitudinal and counter-facade tie-rods inserted in the masonry walls for seismic purposes.

### 3.3. Buttresses

Among the most widely used strengthening techniques aimed at resisting and counter-balancing seismic forces in masonry buildings, buttresses are worthy of being mentioned. In fact, their use was rather common in ancient churches. These elements were conceived as massive structural counterforts and were placed at critical locations, such as the corner or mid-span of the aisle wall of the churches, to counter the lateral thrust arising from the vaulted roofing system of the nave(s) (Figure 17a). According to the "line of thrust" theory [73], the path of the resulting compressive forces passing within a retaining wall or an arch must lie entirely inside the wall thickness of the masonry to avoid the formation of

hinges; hence, the buttresses constituted one of the best solutions to provide massive local additions without increasing the area of the lateral walls (Figure 17b).

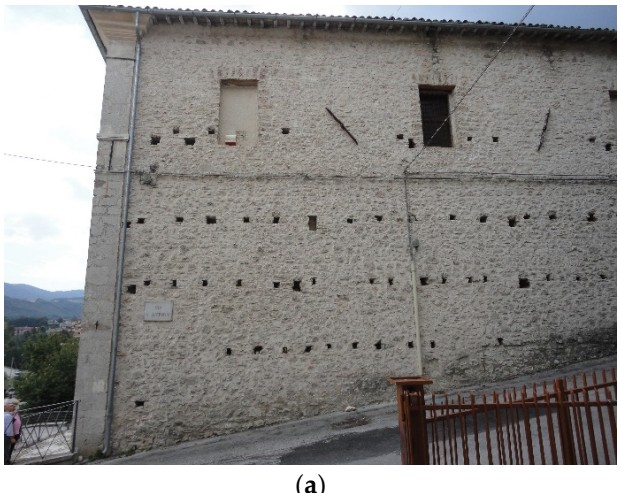 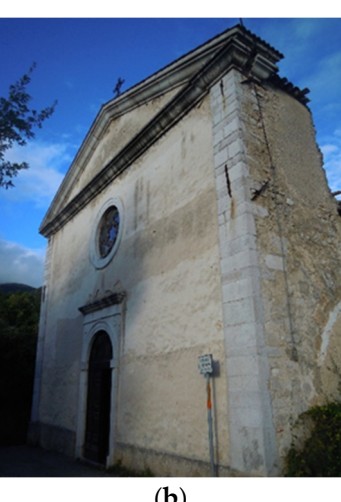

(**a**)                                                                                       (**b**)

**Figure 16.** Metallic tie-rods: (**a**) transversal ties in S. Antonio church in Cascia; (**b**) longitudinal ties in S. Antonio church in Norcia.

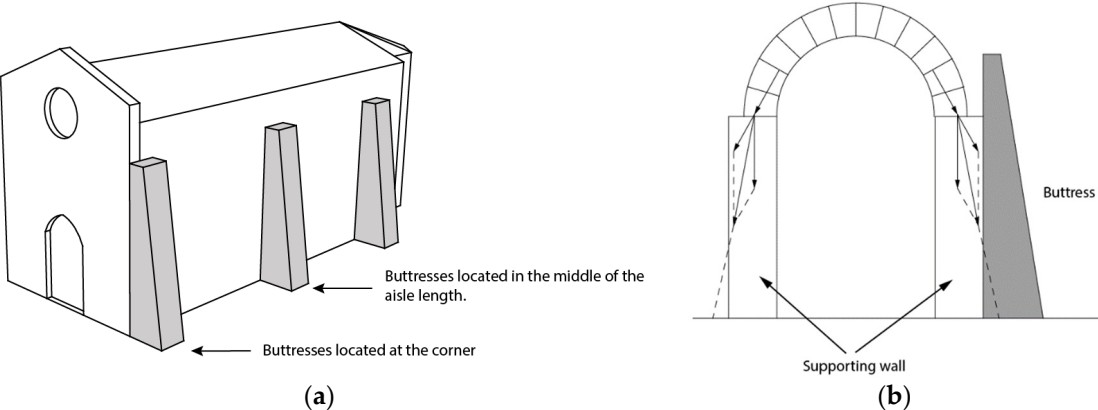

(**a**)                                                                                       (**b**)

**Figure 17.** (**a**) Possible locations of buttresses in church; (**b**) theory of "line of thrust".

These elements, already in use during the Romanesque period, progressively evolved over time, becoming indispensable in the Gothic architecture due to the increasing ambition for lighter structures, lifting the vaults higher and higher [80]. Yet, despite the advantages inherent in their use, the construction/addition of buttresses in churches with several naves did not represent a desirable solution, given the necessity for these elements to be in contact with the wall at ground level, thus obstructing the space of the aisles. Hence, a new solution consisting of "flying buttresses" was conceived to convey to the ground the thrusts of the vaulted ceilings of churches (Figure 18).

This optimal buttress system was composed of two parts: a large external masonry pier rising from the outer wall of the aisle, and an arch element bridging the span between this pier and the upper portion of the central nave wall.

Although this system was seldomly applied in the churches of Central Italy, it is possible to observe one example in the famous single nave church of Santa Chiara in Assisi. The functional efficiency of flying buttresses boosted their use as static supporting systems in many important churches of the time. Yet, their employment was less notable in ordinary churches.

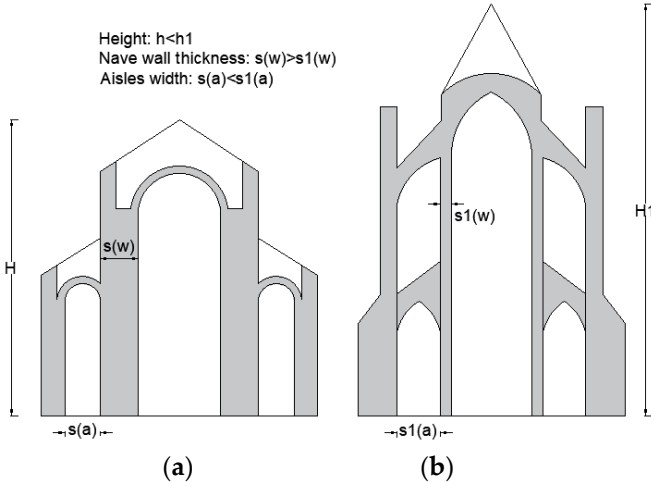

**Figure 18.** Spatial evolution. (**a**) Romanesque church section with semicircular arches and massive walls; (**b**) Gothic church section with rampant arches and slender nave walls.

With reference to the churches of Central Italy taken into consideration within the scope of the present work, two main types of buttresses can be distinguished: the oldest ones, built by widening outwards the base of masonry walls to contrast the slope of the ground as well as prevent out-of-plane mechanisms, as observed in the church of Santa Maria Argentea in Norcia (Figure 19a); and the most common and "recent" ones, namely buttresses inserted with purely anti-seismic purposes, after the expansion of the existing structure. For example, this is the case of the San Francesco church in Terni, Umbria, where unconventional semi-cylindrical buttresses were built against the lateral walls following the enlargement of the church (Figure 19b), or the case of San Agostino church in Cascia, in the same region, where triangular retaining buttresses projecting from the lateral walls were probably added following the increase in height of the structure (Figure 19c), as the different masonry textures suggest.

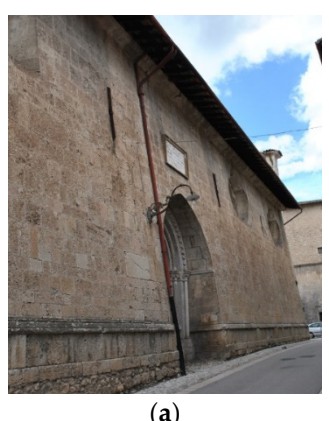
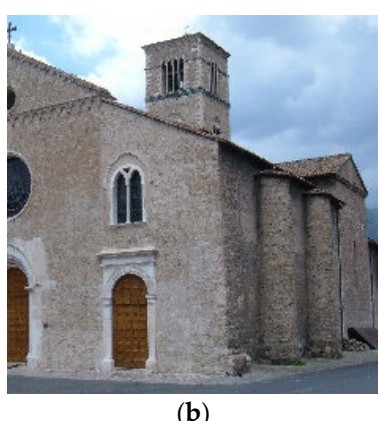
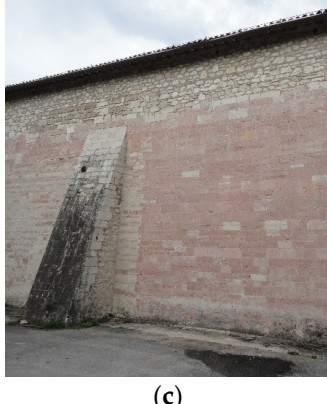

(**a**) (**b**) (**c**)

**Figure 19.** Examples of different types of buttresses: (**a**) buttresses obtained by widening outwards the masonry walls at the ground level in S. Maria Argentea church, Norcia; (**b**) circular buttresses inserted after the expansion of San Francesco Church, Terni; (**c**) classic "triangular" buttresses in San Agostino Church, Cascia.

The main problem with the latter type of buttresses is their connection to the original masonry. In fact, if a buttress is not well anchored to the pre-existing masonry, in the event of facade overturning, its complete detachment could occur, making it completely ineffective.

*3.4. Interventions on the Roof*

As mentioned in Section 2, the prevailing roofing system adopted in many masonry churches in Central Italy consisted of wooden trusses. Apart from being very light, thanks to the presence of a horizontal tie beam, trusses take on all the outward thrust transferred by the rafters, allowing the transmitting of only vertical loads to the underlying masonry walls. To avoid unthreading, tie beams were restrained to the supporting wall through a wooden stake (Figure 20a). This specific truss system, very common in Abruzzo, is known as *"capriata impalettata"* (stacked truss).

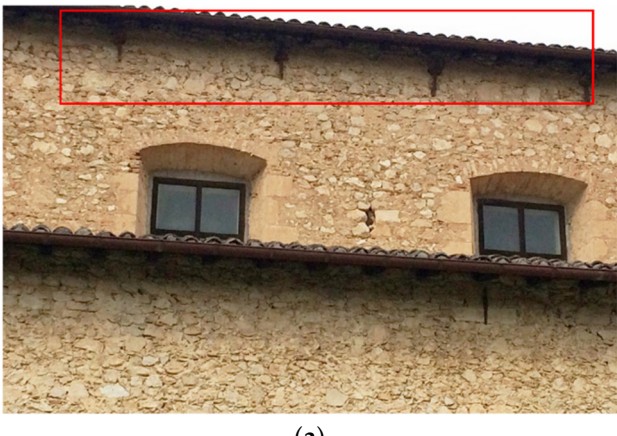

(**a**)

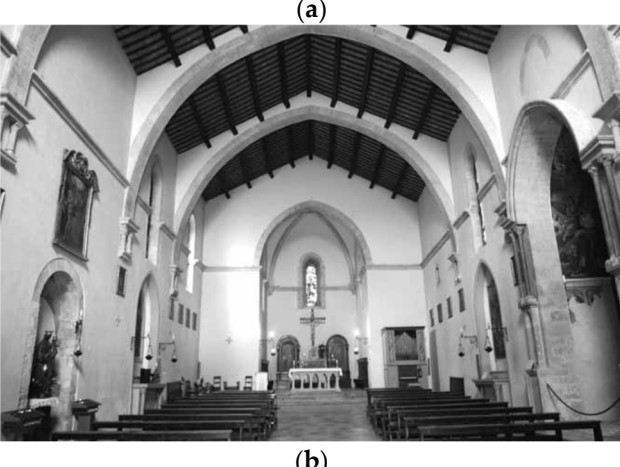

(**b**)

**Figure 20.** Historical presidia in coverage: (**a**) stacked truss in Santa Maria Extra Moenia church, Petogna (AQ); (**b**) diaphragm arches in San Giovanni church, Gubbio (PG) (photo from [81]).

In the area of Umbria and Lazio it is common to find so-called diaphragm arches. This is a wooden roofing system in which the function of struts is replaced by arches [81] (Figure 20b).

In the 20th century, with the advent of new materials and building techniques, traditional anti-seismic presidia were progressively replaced by more "sophisticated" retrofitting systems, with the aim of upgrading the box-like behavior of unreinforced masonry constructions and boosting their structural integrity. In fact, during the post-earthquake reconstructions carried out over the last century, many timber trusses and light roofing systems were dismantled and substituted with heavy reinforced concrete (RC) roofs (Figure 21a) [14]. Moreover, to improve the wall-to-roof connections and the transmission of horizontal fores, RC ring-beams were frequently inserted all along the perimeter of the churches (Figure 21b), under the roof structure, accommodating them into longitudinal cuts excavated inside the masonry walls for more than half of their thickness and leaving only a small portion of the cladding stones for aesthetics purposes. It was a practice widely used for floor slabs in multi-story buildings to make them resistant to seismic actions.

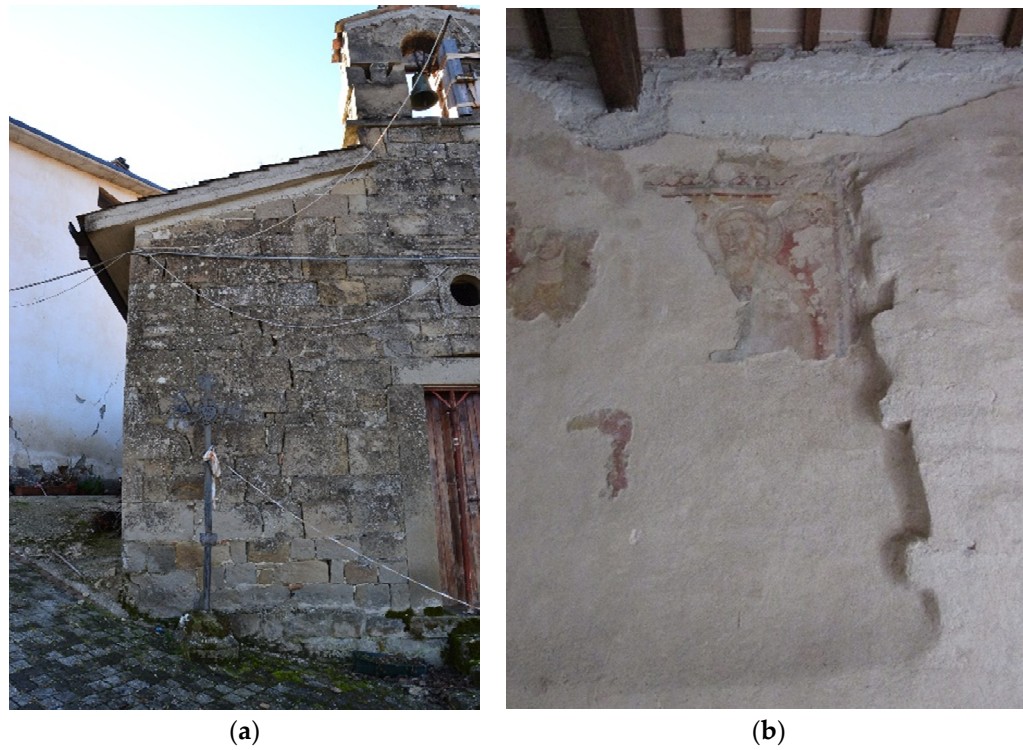

<center>(<b>a</b>)                                            (<b>b</b>)</center>

**Figure 21.** Example of interventions on the roof: (**a**) heavy concrete roof of Madonna del Carmine Church, Cortino (TE); (**b**) rigid beams inserted in the walls of the S. Martino Church, Norcia (PG).

Though, in most cases, such additions were harmful for CH buildings and churches, not only because they caused a reduction of the cross-sectional area of the underlying masonry walls, adversely affecting the overall seismic behavior and increasing the vulnerability to out-of-plane failures when badly connected, but also because they produced a significant increase of the seismic masses at the top of the structure, contributing to higher earthquake-induced inertia forces. Whatever the mode of construction, it is known that RC elements feature a remarkable bearing capacity; however, they are too heavy and stiff to be combined with URM construction systems. Indeed, the different properties of concrete and masonry can lead to diverse vibration responses and result in negative effects when the building is subjected to seismic loading, generating high local stresses in the masonry elements. Moreover, RC-based reinforcement techniques are irreversible interventions, and for this reason, they are nowadays not recommended as a seismic improvement technique in built cultural heritage. A deeper insight into these aspects will be given in the Section 4.

Today the new approach to conservation dictates that seismic improvements have to be achieved without invasive interventions and respecting the "authenticity" of the construction, both in terms of materials and geometries [82]. Accordingly, smart retrofitting strategies based on the use of dissipative and deformable roof structures (for example, with steel connections, light and reversible interventions) have been fruitfully proposed in the last few years [43].

## 4. Assessment of the Seismic Interventions

In this Section, the seismic behavior of masonry churches is analyzed by subdividing them into macroelements, i.e., architectural portions with similar seismic responses. The evaluation of the macroelements and associated collapse mechanisms started with the Friuli earthquake of 1976 [15] and culminated with the definition of the A-DC survey form for churches in 2013 [48]. By exploiting these forms, the post-emergency activities carried out on cultural heritage buildings after the 2016 seismic sequence [83] enabled assessment of the earthquake-induced damage surveyed in the investigated sample of masonry churches in relation to the 28 possible collapse mechanisms defined by the Italian "Guidelines for

the evaluation and risk reduction of Cultural Heritage" [84]. Indeed, the compilation of the form allowed, for each collapse mechanism, to rate the observed damage with a grade from 0 (no damage) to 5 (collapse), according to the damage criteria defined by Grünthal [85] (introduced for buildings, but transposed for single macro elements in several subsequent research works [24]), and to infer therefrom an overall damage index ($i_d$) [86], making possible a statistical evaluation of the most common damage mechanisms that occurred in various types of churches.

The practical goal of the A-DC form is the rapid evaluation of the usability of the churches [87,88] as well as the definition of emergency measures and future costs of restoration; "usable" indicates a church without structural damage; "usable with countermeasures" refers to churches that need protection measures before being opened; "partially usable" refers to churches that are partially accessible only; "temporarily unusable" indicates churches whose survey is not complete, and further inspections are needed to take definite decisions; "unusable for external risk" indicates the presence of external unsafe elements that threaten the safety of the church; "unusable" refers to churches whose structural stability is compromised. Figure 22 shows the usability assessment outcome of the sample of churches presented in the introduction. As mentioned above, the sample consists of churches directly surveyed by the authors plus additional churches catalogued within the Da.do database.

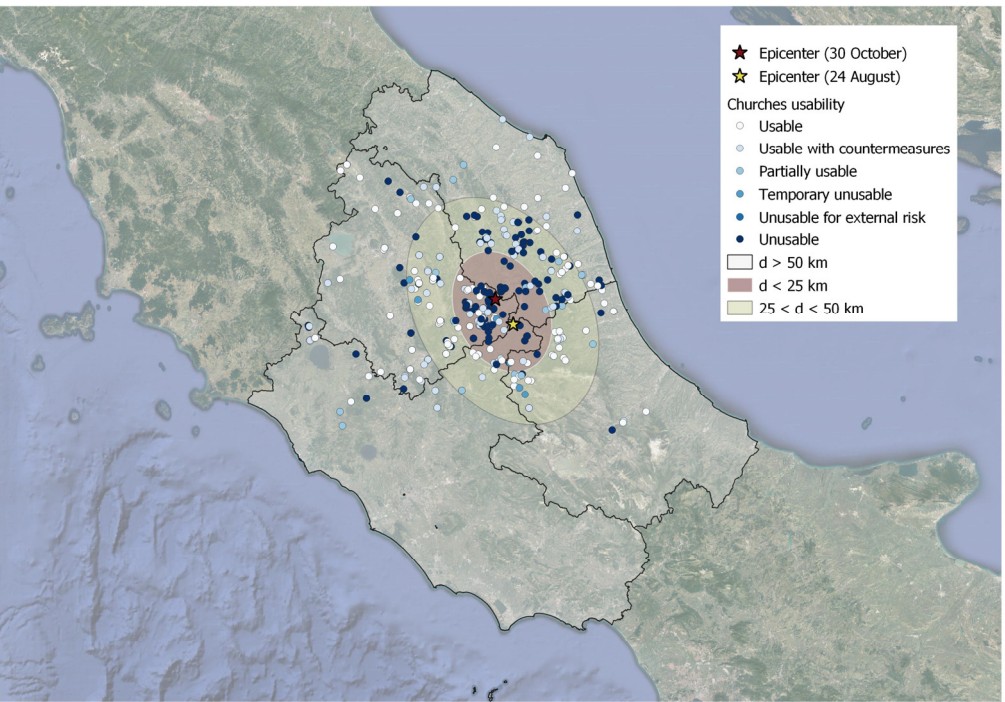

**Figure 22.** Usability of the sample of over 600 churches in Central Italy after the 2016 earthquake according to the A-DC form (source: Da.Do database). The colored areas identify the distances of the churches from the epicenters of 24 August and 30 October 2016, as shown in the legend.

The form also proposes suitable safety measures, depending on the type of identified mechanism. As for Central Italy, propping was the most used temporary counter measure to inhibit the overturning of facades; very often, traditional propping systems were integrated with the use of steel cables or polyester bands, which proved suitable for counteracting the overturning of the cracked facades, in particular those ones detached by the orthogonal walls, without obstructing the streets around the buildings [89].

### 4.1. Central Italy Seismic Sequence

The area mostly affected by the 2016–2017 earthquakes involves the territories of Abruzzo, Lazio, Umbria and Marche. Reliable earthquake characterization is important

to better understand the seismic effects on the territory [90]. Therefore, a summary of the most important features of the main shocks of the Central Italy seismic sequence is given below (data retrieved from INGV, http://terremoti.ingv.it, accessed on 26 November 2022):

- 1st shock—24 August 2016: epicenter in Amatrice, Mw = 6.0;
- 2nd shock—26 October 2016: epicenter in Castelsantangelo sul Nera, Mw = 5.4
- 3rd shock—26 October 2016: epicenter in Visso, Mw = 5.9
- 4th shock—30 October 2016: epicenter in Norcia, Mw = 6.5
- 5th shock on 18 January 2017: epicenter in Capitigliano, Mw = 5.5

One month after the first shock in August, numerous groups from different Italian universities, through an agreement of DPC, MiBACT and ReLUIS, carried out an assessment of the usability and damage of churches through the A-DC form. In this work, the maximum intensity measure registered during the whole seismic sequence before the inspection date of the specific church was considered. Because most of the surveys used for the database by the authors were conducted after the first and fourth shocks, these were chosen as the reference earthquakes for the statistical analyses, presented next. In this regard, Figure 23 shows the peak ground shaking provided by the Da.Do web gis platform, which merges the new implementation of ShakeMap at INGV [91] and the damage data collection [92].

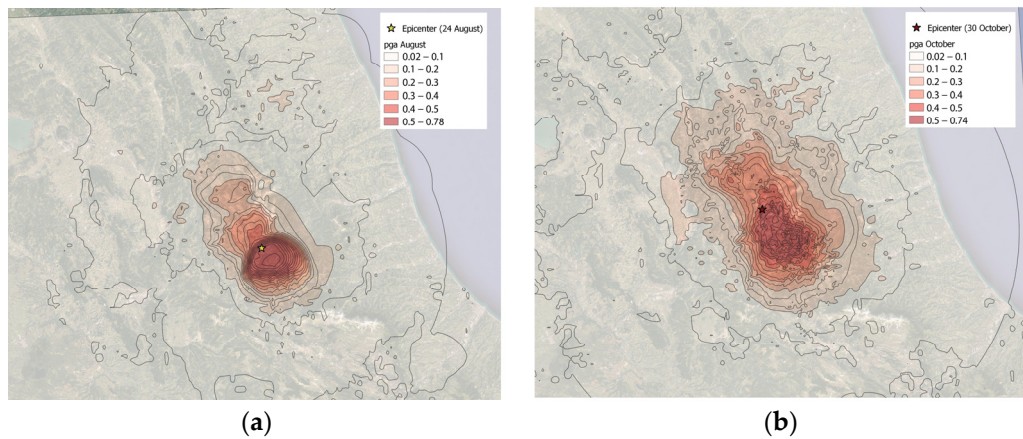

(**a**)　　　　　　　　　　　　　　　　(**b**)

**Figure 23.** ShakeMaps for (**a**) 24 August 2016 and (**b**) 30 October 2016 earthquakes in Pga (g).

The effects of the 2016 earthquakes on the investigated sample of more than 600 churches were carefully analyzed on the basis of the forms completed during onsite inspections and the photographic documentation acquired during the aforementioned reconnaissance activities, allowing a critical assessment of the strengthening and repairing interventions executed in the years preceding the event, whenever these were visible. The outcome of this evaluation process is detailed in the following, in relation to the major interventions observed in the sample of churches: ties rods and ring beams. In fact, among the types of interventions discussed in Section 3, these were the most prevalent, thereby allowing a reliable statistical evaluation of their effectiveness.

### 4.2. Effective vs. Ineffective Interventions

In order to proceed with the evaluation process, churches were grouped by structural characteristics and epicentral distances. The first group of churches is located at distance d < 25 km from the two major epicenters of 24 August and 30 October, which are considered the most influential in terms of magnitude (6.0 Mw and 6.5 Mw, respectively); the second group includes churches located at distance 25 km < d < 50 km, while the third group includes churches sited at distance d > 50 km (Figure 23).

Comparing the results in terms of unusability, the data show that, independently of the epicentral distance, churches with ring beams always feature higher percentages of unusability (hence greater severity of seismic damage) than churches with longitudinal

and/or transversal tie rods (Figure 24). For completeness, Table 2 shows the usability percentages obtained for each group of churches.

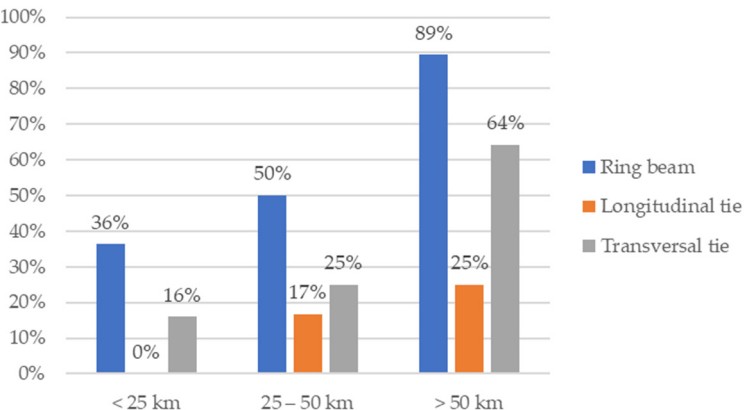

**Figure 24.** Unusability of the sample of churches according to the distance from the epicenter and to the type of intervention.

**Table 2.** Usability percentages divided by distances to epicenters and types of intervention. In red, the prevailing usability for each distance and intervention.

|  | Ring Beam | | | Longitudinal Tie | | | Trasversal Tie | | |
| --- | --- | --- | --- | --- | --- | --- | --- | --- | --- |
|  | >50 km | 25–50 km | <25 km | >50 km | 25–50 km | <25 km | >50 km | 25–50 km | <25 km |
| Unusable | 36% | 50% | 89% | 0% | 17% | 25% | 16% | 25% | 64% |
| Partially unusable | 9% | 6% | 5% | 40% | 17% | 15% | 20% | 13% | 0% |
| Accessible with provision | 27% | 13% | 5% | 20% | 17% | 17% | 24% | 29% | 0% |
| Temporarily unusable | 0% | 0% | 0% | 0% | 0% | 5% | 0% | 0% | 7% |
| Accessible | 27% | 31% | 0% | 40% | 50% | 35% | 40% | 33% | 21% |
| Unsafe external dangers | 0% | 0% | 0% | 0% | 0% | 3% | 0% | 0% | 7% |

To better understand the impact of the different interventions on the unusability of the buildings, the post-earthquake conditions of two churches (San Giorgio and San Martino), similar in size and distance from the epicenters (10 km distant from the August epicenter and 25 km from the October one), were compared. Both churches are located in Amatrice, one of the Italian towns greatly affected by the 2016 earthquake [93].

Built in the 19th century, San Giorgio church is a single-nave structure made of plastered irregular masonry, with a gabled facade and vaulted ceiling. To reduce the overall thickness of the vaults and lighten the weight on the underlying walls, bricks were laid flatwise ("*in folio*" vaults). The main intervention carried out in this church over the past years involved the addition of metallic tie-rods at approximately one-third the distance from the impost of the arch in order to retain possible outward movements of the lateral walls. Despite the significant seismic damage occurring in the vaults, with typical diagonal cracks [94] passing through pendentive, the structure remained intact (Figure 25a,b).

By contrast, the church of San Martino, a single-nave stonemasonry structure with roof trusses built in the 15th century, was not able to withstand the seismic sequence of 2016 without being severely damaged. An important intervention was carried out around 1980, with the insertion of RC ring-beams along the perimeter of the church, at the roof level. The inadequate connections with the underlying unreinforced masonry walls, together with the increase in mass originated by the addition of these elements, led to severe collapses. The close-up of Figure 26 clearly shows the lack of anchoring between the ring-beam and the masonry wall below, which favored sliding. It is also important to stress the poor quality of the masonry on which the intervention was made as well as the inappropriate design of the rigid beam, whose width does not equal the wall thickness, thus being ineffective.

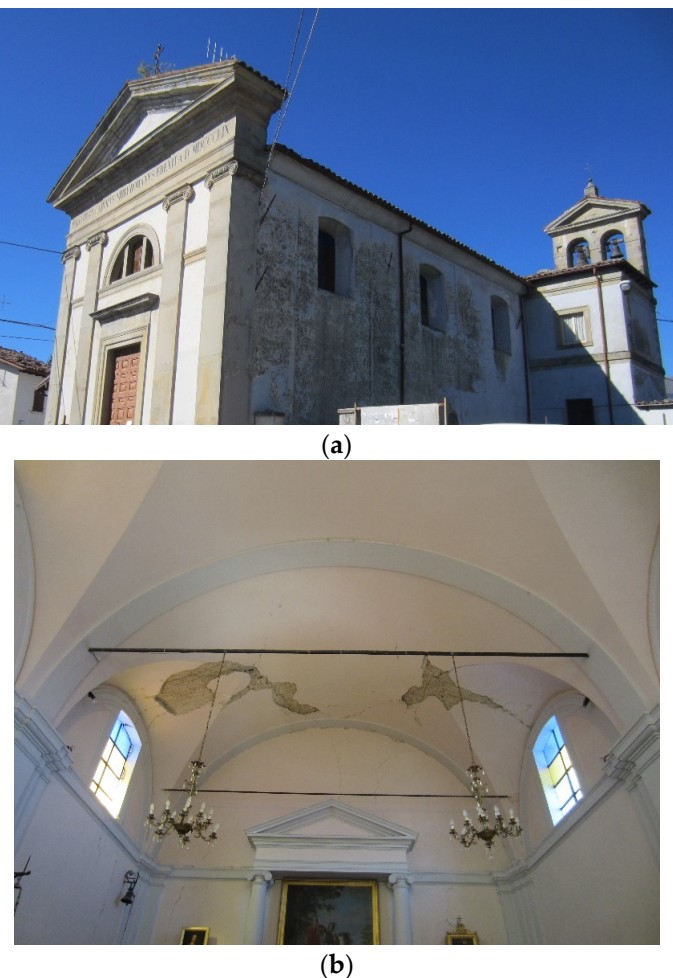

**(a)**

**(b)**

**Figure 25.** San Giorgio church in Amatrice: (**a**) external view; (**b**) detail of the "in folio" vault and transversal tie rods.

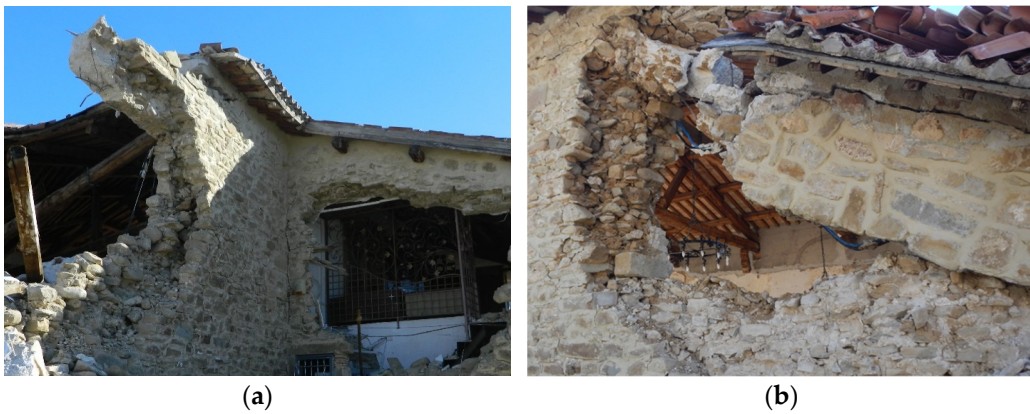

           **(a)**                                             **(b)**

**Figure 26.** San Martino Church in Amatrice: (**a**,**b**) adverse effect of the RC ring-beam.

To further investigate the influence of seismic retrofitting measures on the analyzed sample of churches, a valuable comparison was made, taking into account three specific collapse mechanisms strongly influenced by the presence of tie-rods or ring beams: mechanism 1—overturning of the facade, mechanism 2—overturning of the top of the facade, mechanism 5—transversal response of the nave. For each mechanism, the damage level recorded in the churches retrofitted with ties (either deployed transversally or longitudi-

nally) was weighed against the damage level suffered by churches reinforced with rigid RC beams.

The first correlation aimed at analyzing the damage caused by the activation of the facade overturning mechanism in churches retrofitted with longitudinal tie-rods versus those retrofitted with RC ring beams. As observed in Figure 27a, only 5% of the churches with tie-rods developed severe damage levels (d3-d4-d5) for the considered collapse mechanism. This percentage increased in presence of rigid RC ring-beams only, increasing to 15% for a damage level of 5, i.e., collapse.

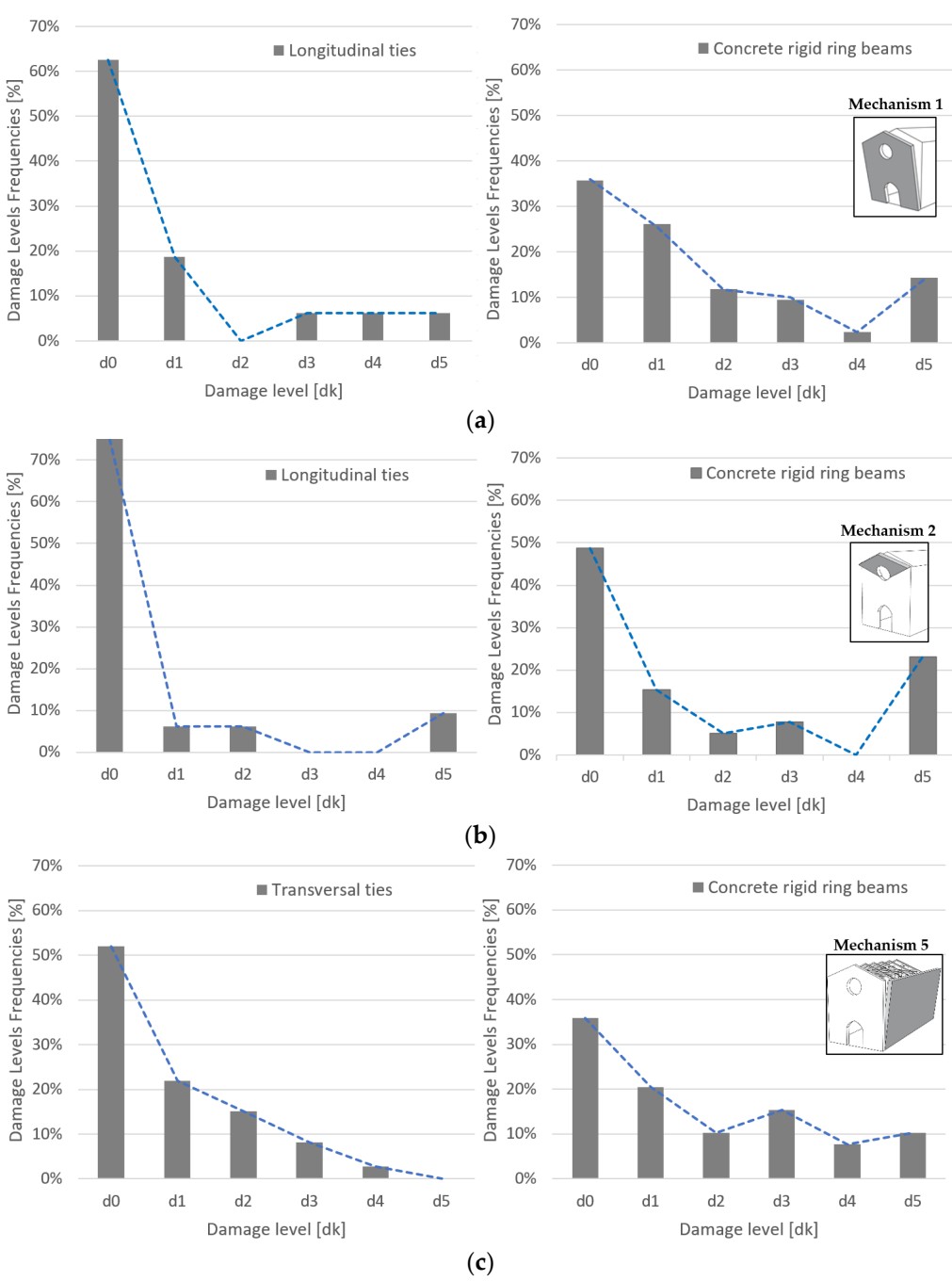

**Figure 27.** Comparison between damage on mechanism 1 (**a**) and mechanism 2 (**b**) in the presence of longitudinal tie rods and RC ring beams; (**c**) comparison between damage on mechanism 5 in the presence of transversal tie rods and RC ring beams.

The second comparison was made by examining, for the same churches, a different collapse mechanism, namely the overturing of the top of the facade (Figure 27b). Although the damage trend is similar to mechanism 1, an evident increase in the number of churches suffering from higher damage levels is recorded in the presence of RC ring-beams. According to the results, this collapse mechanism is likely due to the lack of connections between elements and the multi-layer masonry with an inner core made of poor quality rounded stones and with an external masonry made of regular stones (Figure 26).

As a final comparison, the damage level resulting from the activation of the mechanism involving the transverse response of the nave was analyzed. In this case, the severity of damage drops in the presence of transversal tie-rods, whereas a higher percentage of churches showing higher levels of damage is found in the presence of RC ring-beams (Figure 27c). Interventions of this type are not a priori incorrect, but for the investigated churches, i.e., masonry structures with Ministry-imposed limitations and macro-element behavior, they were not effective.

The results show indeed that the use of stiff roofs, especially if not effectively connected to the underlying load-bearing masonry walls, does not improve the seismic response of the structure; on the contrary, it may lead to unexpected and dangerous seismic behaviors. Lighter steel horizontal diaphragms, properly connected to the vertical resisting system, as well as restraining tie rods between orthogonal walls can provide valid alternatives to reinforced concrete elements in order to foster a better seismic performance. In this regard, interventions against masonry crumbling are also fundamental. Generally, as for heritage structures of this kind, seismic retrofitting interventions should primarily aim at improving the global and local response of the masonry against horizontal actions, which means providing adequate connections between the different macro-elements composing the building to avoid global out-of-plane mechanisms, inserting interlocking elements between masonry leaves to prevent local wall disintegration and injecting compatible grouts into masonry to increase resistance against deformation and improve material cohesion. When dealing with built heritage, reversibility and compatibility of the materials must be always guaranteed.

### 4.3. Aftershock Damage Due to Lack of Prompt Intervention

An aspect often neglected but worthy of consideration is related to the incommensurable damage caused to churches by the lack of prompt interventions in the aftermath of strong seismic events, particularly in the presence of relevant aftershocks.

Partial or complete loss of the heritage may occur if adequate propping systems are not installed in a timely manner, as was the case in the church of Santa Giusta, 7 km away from Amatrice.

Santa Giusta is a single nave church with a lunette barrel vault. As emerged from the analysis of the A-DC form compiled in September 2016, the bell tower was raised in 1970 (Figure 28a,b). Although the absence of metal ties between orthogonal walls was a constant threat to the structural integrity, the bell tower of Santa Giusta church was able to withstand the major seismic shocks of the 2016 Central Italy earthquake. Though, due to the missing installation of temporary propping systems to support unstable damaged parts, the seismic event of January 2017 caused the complete collapse of the tower (Figure 28c).

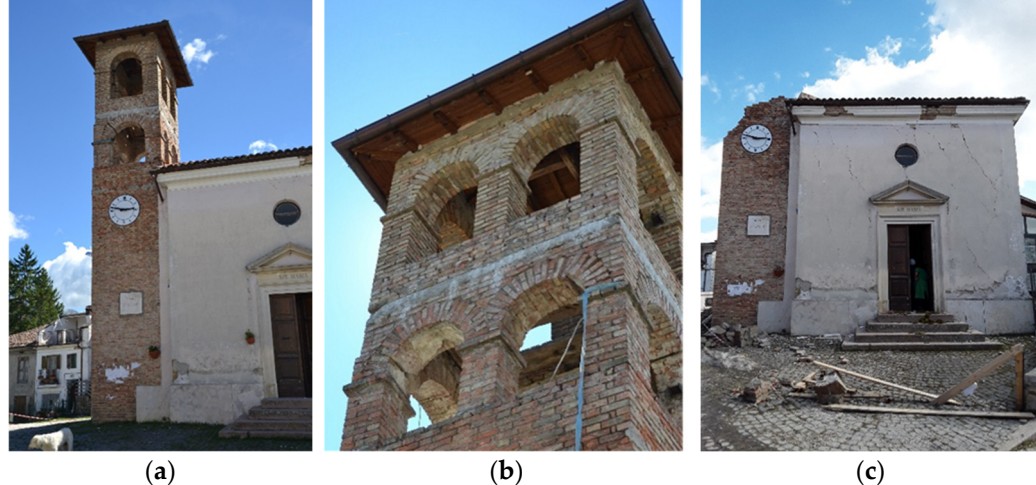

|       |       |       |
| :---: | :---: | :---: |
| (**a**) | (**b**) | (**c**) |

**Figure 28.** Exterior damage observed for the Santa Giusta church (Amatrice, Italy): (**a**) overview and (**b**) close-up of the damaged bell tower; (**c**) the bell tower after the collapse.

An analogous comparison was made considering the barrel vault in "*camorcanna*", a vaulted system with a light wooden structure particularly widespread in Central Italy [69].

The first seismic events damaged the vault, and disconnections were observed by the technicians during the survey (Figure 29a), hence leading to a recommendation of the adoption of safety measures. Unfortunately, the lack of a prompt intervention led the vault to collapse during the following 5.5 magnitude earthquake of January 2017 (Figure 29b).

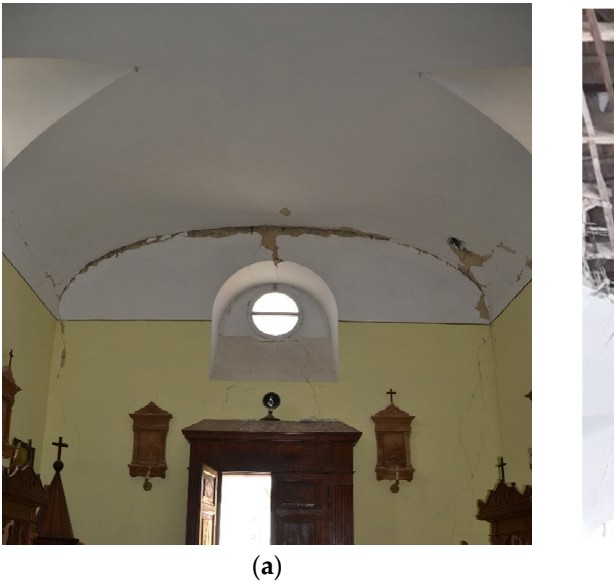
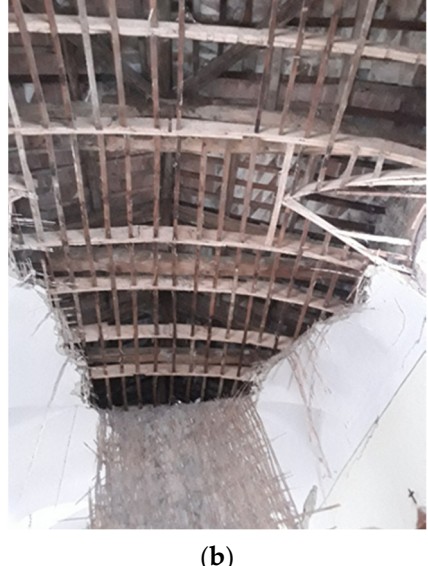

|       |       |
| :---: | :---: |
| (**a**) | (**b**) |

**Figure 29.** Internal view of the church of Santa Giusta in Amatrice: (**a**) lunette vault in "*camorcanna*" (September 2016); (**b**) collapse of the vault (January 2017).

### 4.4. Damage Due to Historical Stratifications

The structural discontinuities originating from subsequent construction phases and a posteriori additions can create strong vulnerabilities, also leading to the partial or complete loss of the heritage structure. Building expansions, if not well connected to the existing structure, always generate points of discontinuity. As an example, Figure 30a shows a vertical crack separating the original wall of the church of San Rocco in Pietracamela (Abruzzo region) from the wall of the sacristy added in a second phase, thus lacking a proper interlocking with the pre-existing structure. Similar damage is visible in the Abruzzo church of S. Maria Immacolata in Isola del Gran Sasso, where the crack occurring

at the interface between the pre-existing arch and the walled-up doorway called for urgent remedial measures to prevent possible collapse (Figure 30b). Recognizing the vulnerability of heritage structures in relation to their historical stratifications is of utmost importance to prioritize both rescue operations and propping interventions during the earthquake emergency phase, thus avoiding greater damage and inestimable losses.

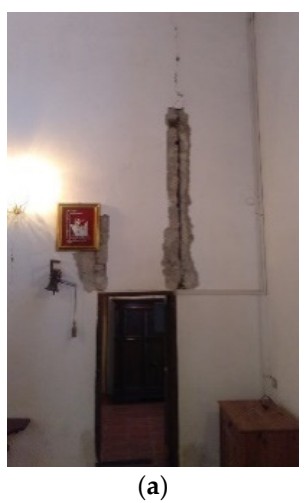
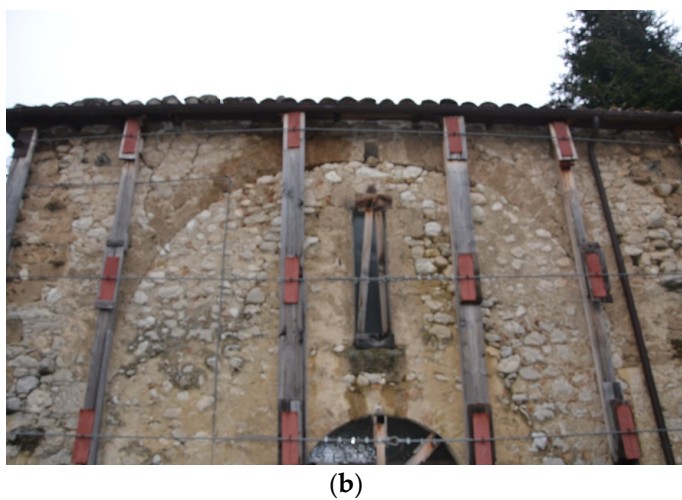

(**a**)  (**b**)

**Figure 30.** Masonry discontinuity: (**a**) San Rocco Church, Pietracamela; (**b**) S. Maria Immacolata Church, Isola Del Gran Sasso.

## 5. Conclusions

Earthquakes have always marked the history of the affected territories and of their cultural heritage. Indeed, many of these events have not only caused losses of human lives but have also influenced the economy and the depopulation of several areas. Building reconstructions can be costly and challenging but are also a chance to explore new techniques, for both stylistic and structural modernization.

In this paper, an overview of the structural features—relevant from a seismic viewpoint—of a significant sample of 600 churches located in Central Italy was presented, following the damage reconnaissance carried out in 2016 and 2017. The main post-earthquake interventions applied in the past to these heritage churches have been extensively described and their effectiveness discussed through comparative statistical analyses. The main fragilities that have threatened, and continue to threaten, this type of construction have been also identified.

In the light of the findings from the post-earthquake survey conducted on the aforementioned sample of churches and of the extensive analyses performed hitherto, the following conclusions can be drawn:

- The structural features of the surveyed churches, which soundly affected their seismic response, are often the result of past seismic events and subsequent post-earthquake reconstruction processes, whose knowledge is fundamental to provide reliable predictions of future damage scenarios;
- The seismic damage related to out-of-plane mechanisms was sufficiently reduced by the presence of transversal and longitudinal ties; conversely, churches with heavy RC roofs and ring beams, especially if placed on a poor-quality masonry, led to the worst seismic behavior, leading to 89% unusable churches in the area located at distance d < 25 km from the epicenter;
- Additions to the original plant represented a further source of vulnerability against horizontal actions;
- The lack of prompt temporary interventions after the first shock increased the proneness of the churches to undergo cumulative permanent damage during the aftershocks, which in some cases led to partial or complete failures.

The present work is not intended to be exhaustive, but it provides the first results from months of data collection and years of data storage, hierarchical classification and comparative analyses aimed at extracting valuable information to feed seismic fragility curves and predict future potential damage scenarios. In this sense, this work represents an important baseline reference for more accurate large-scale vulnerability assessments at the supra-regional scale.

It is important to highlight that the variables affecting the severity of the seismic damage observed in the analyzed sample of churches are not only related to the magnitude of the seismic shake, the distance of the structure from the epicenter and the type of retrofitting/strengthening intervention. Indeed, local site effects can play a major role, ultimately altering the expected damage scenario. This aspect adds further complexity to the study and is currently the subject of extensive research by the authors.

The obtained results will be ultimately exploited to identify archetypes of churches, namely church models featuring analogous (expected) seismic behavior, based on their typological characteristics and the interventions undertaken in the past. The definition of archetypes will enable in-depth empirical and analytical studies in order to quantify the seismic capacity improvement associated with different retrofitting solutions and to compare the costs and benefits of each intervention in historic churches for decision-making support at the governmental level.

**Author Contributions:** Conceptualization, G.C. and M.G.M.; methodology, G.C. and M.G.M.; validation, G.C.; investigation, G.C.; resources, C.V.; writing—original draft preparation, G.C.; writing—review and editing, G.B. and M.G.M.; supervision, G.B. and M.G.M.; funding acquisition, G.B. All authors have read and agreed to the published version of the manuscript.

**Funding:** The present study was carried out within the framework of the Reluis 2019–2021 Italian Research Project, funded by the Italian Civil Protection, as well as within the framework of the National Operational Programme on Research and Innovation (Attraction and International Mobility) PON-AIM 2014-2020 Line 2, co-financed by the European Social Fund and by the National Rotation Fund.

**Institutional Review Board Statement:** Not applicable.

**Informed Consent Statement:** Not applicable.

**Data Availability Statement:** Data supporting the study are available from the corresponding author upon reasonable request.

**Acknowledgments:** The authors would like to thank the colleagues who participated in the surveys of churches during the emergency phases following the 2016 Central Italy earthquake.

**Conflicts of Interest:** The authors declare no conflict of interest.

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
