# Peer review of "An Overview of the Historical Retrofitting Interventions on Churches in Central Italy"

_applsci, doi:10.3390/app13010040_

Round 1
Reviewer 1 Report
The study is very interesting and important, but unfortunately it lucks quantitative comparisons. Need to provide at least with magnitude and distances of earthquakes to churches and possibly intensity of shaking. Otherwise it is not clear how strong shaking was in a specific area. May be it was effect of higher shaking and not style of anti-seismic efforts.
My comments are also in attached PDF.

Reviewer 2 Report
This a new attempt to better understand the behaviour of churches in Central Italy due to earthquake sequences. I have some minor points which could improve the quality of work as follows:
1- More information about the sequences used in the paper, could be put in the text including peak ground motions parameters and acceleration spectra as well.
2- Limitation of the results could be mentioned clearly.
3- Literature review of paper could be improved including some new works on earthquake sequences, for example
Time–frequency analysis of the 2012 double earthquakes records in North-West of Iran, published in Bulletin of earthquake engineering 12 (2), 585-606
Reviewer 3 Report
The paper lacks of originality. It consists of an overview with no novel contributions of the 2016 earthquake: 6 years ago! The reviewer considere this paper more suitable for a conference than to a journal.
Reviewer 4 Report
The text is well written, and the information is presented in a clear, if somewhat long, manner. The results of the survey performed in this study are also clearly stated and summarized, indicating what type of problems can be related with the different types of interventions.
There are only two comments that I could make on this manuscript:
1. I may find that the methods used to evaluate the different intervention mechanisms are not that clearly described and could be improved.
2. While their goal is evaluating the effectiveness of the different intervention techniques, I think the paper would benefit from the authors giving some recommendations as for the direction in which the preservation of these heritage churches should follow.
Even if these details can be improved, I believe the article is ready for publication in its present form.
Additionally, I have some small comments on a couple of figures. It isn't necessary to attend them, but I believe they could help the reader in understanding the figures.
Figure 2. I understand the map is just meant to give an idea that there is a higher church density in the northern part of central Italy (more or less the Papal States). However, how large are municipalities in Italy? I'm guessing size varies a lot between them, so how can the reader know that the map represents a higher church density instead of just smaller municipalities in this area?
Figure 4. Please specify in the figure caption what the percentage shown on top of each data point is indicating. Also, is this percentage for all of Italy, or just for the about 600 churches studied in this work?
Figure 7. I'm guessing it's the same church, just in one image the front is shown, and in the other the back. I know it's not important for the discussion of the article, but if you're going to provide an image to compare the changes, it'd be better to have a similar view. Wikipedia (the source of Figure 7a) actually has an image of the interior in the same direction as the picture from before 1934, just that in this picture the roof is barely visible.
Round 2
Reviewer 3 Report
The authors need to describe the novelties of the paper in order to support the originality of what they are presenting.
Also, the due reference are necessary to support the study.
Conclusion needs to be more propositive.
